# AdapTable: Test-Time Adaptation for Tabular Data via Shift-Aware Uncertainty Calibrator and Label Distribution Handler

## Abstract

In real-world applications, tabular data often suffer from distribution shifts due to their widespread and abundant nature, leading to erroneous predictions of pretrained machine learning models. However, addressing such distribution shifts in the tabular domain has been relatively underexplored due to unique challenges such as varying attributes and dataset sizes, as well as the limited representation learning capabilities of deep learning models for tabular data. Particularly, with the recent promising paradigm of test-time adaptation (TTA), where we adapt the off-the-shelf model to the unlabeled target domain during the inference phase without accessing the source domain, we observe that directly adopting commonly used TTA methods from other domains often leads to model collapse. We systematically explore challenges in tabular data test-time adaptation, including skewed entropy, complex latent space decision boundaries, confidence calibration issues with both overconfident and under-confident, and model bias towards source label distributions along with class imbalances. Based on these insights, we introduce *AdapTable*, a novel tabular test-time adaptation method that directly modifies output probabilities by estimating target label distributions and adjusting initial probabilities based on calibrated uncertainty. Extensive experiments on both natural distribution shifts and synthetic corruptions demonstrate the adaptation efficacy of the proposed method.

## 1 Introduction

Tabular data is one of the most abundant forms of data in the industrial world, covering diverse fields such as healthcare (Johnson et al., 2016; 2021), finance (Shah et al., 2022), and manufacturing (Hein et al., 2017) to name a few. However, due to their ubiquity and vast quantity, tabular data in the wild frequently exhibits *distribution shifts* (Malinin et al., 2021; Üstev et al., 2013). For instance, electronic health record (EHR) data collected from a senior-majority cohort may differ in distribution compared to a junior-majority cohort. Under such data distributions, machine learning models trained on data from the senior cohort may not function as intended on data from the junior cohort at test time. Such distribution shifts pose a significant problem in deploying learned models as they undermine their integrity at test time.

Despite the importance of addressing such prevalent distribution shifts in tabular data, there have been limited studies on domain adaptation for tabular data compared to other domains like computer vision (Zhu et al., 2023; Wang et al., 2021a), natural language processing (Farajian et al., 2017; Dou et al., 2019) and speech processing (Mai & Carson-Berndsen, 2022; Kim et al., 2023). The difficulties of developing domain adaptation strategies for the tabular realm stem from two primary reasons. First, tabular data displays unique and heterogeneous attributes, including both numerical and categorical features across columns, where these attributes exhibit substantial variations. Second, deep tabular learning models often exhibit limitations in their representation learning schemes in tabular data, which impose constraints on domain adaptation methods that rely on utilizing the feature space of the model. These factors are intricately intertwined, making tabular domain adaptation exceptionally challenging.

In parallel, test-time adaptation (TTA) (Wang et al., 2021a; Liu et al., 2021; Niu et al., 2023; Sun et al., 2020; Lim et al., 2023; Mirza et al., 2023; Boudiaf et al., 2022; Zhou et al., 2023; Park et al., 2023)

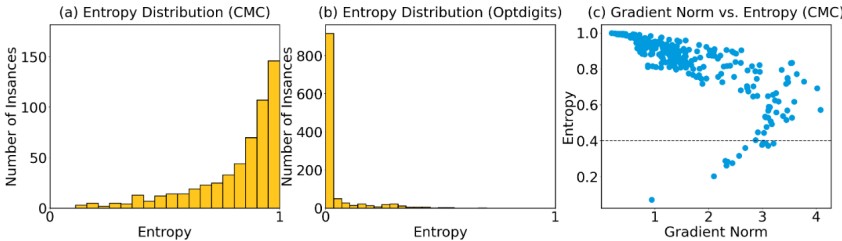

**Figure 1:** Comparison of entropy distribution histograms for tabular (a) and image (b) datasets using the same MLP model architecture, along with a gradient norm vs. entropy plot for the tabular dataset (c).

has been recently proposed as an emerging paradigm in domain adaptation – aiming to adapt the off-the-shelf model to an unlabeled target domain during inference phase without accessing the source domain. TTA methods are appealing due to their suitability in scenarios where accessing source data is impractical, either due to privacy or storage limitations, or in cases where the precise target domain during testing is unknown. In the context of tabular data, where there is a growing demand for TTA, we observe that naively applying existing approaches designed for other domains tends to degrade the performance of the source model in most scenarios. In particular, we systematically examine the failure of entropy minimization-based fully test-time adaptation methods, which are at the forefront of the TTA paradigm for other domains, applied to tabular data. We discover that the prediction entropy of the model consistently exhibits a strong bias toward the under-confident region due to their complex decision boundaries within the latent space, which are known to trigger large gradients within the context of the entropy minimization objective, leading to model collapse. In addition, we find that the predictions of the models are highly biased to the source label distribution as in other domains (Berthelot et al., 2020; Wu et al., 2021; Park et al., 2023). This bias also contributes to the shift of the model towards the major class of the source domain, ultimately leading to model instability.

Motivated by these observations, we propose an **Adap**tation for **Table** (*AdapTable*) framework, which is the first test-time adaptation strategy tailored for the tabular domain. Confirming that using unsupervised objectives during the inference phase to adapt the model is highly unstable for a tabular domain, AdapTable circumvents the need to tune the model parameter using unsupervised objectives and focuses instead on correcting the output distribution. With the findings that the model's prediction probabilities are poorly calibrated and biased to the under-confident area, we propose a shift-aware post-hoc uncertainty calibrator, which predicts per-sample temperature scaling factor to calibrate predictions, in consideration of shift information of columns and their relationships using graph neural networks. Armed with the hypothesis that the source model is highly biased towards the source label distribution, we suggest a label distribution handler, where we estimate the target label distribution of the current batch, and correct the target label distribution grounded in Bayes' theorem. Here, we interpolate the original label distribution and the estimated target label distribution based on the margin calculated by the calibrator. Through extensive experiments, we verify that our method achieves state-of-the-art performance across various datasets including natural distribution shift scenarios, and synthetic corruptions with three representative deep tabular learning architectures.

To summarize, our contribution is threefold:

- We thoroughly analyze the failure of existing entropy minimization-based test-time adaptation strategies on tabular data in terms of the model's entropy distribution, label distribution, confidence calibration, and decision boundary in the latent space.

- With these observations in mind, we propose a test-time adaptation method that directly adjusts output prediction probabilities utilizing shift-aware post-hoc uncertainty calibration and label distribution handler on the tabular domain for the first time, to the best of our knowledge.

- Throughout extensive experiments, we verify that the proposed method achieves state-of-the-art results on different model architectures and out-of-distribution benchmarks including natural distribution shifts and synthetic corruptions.

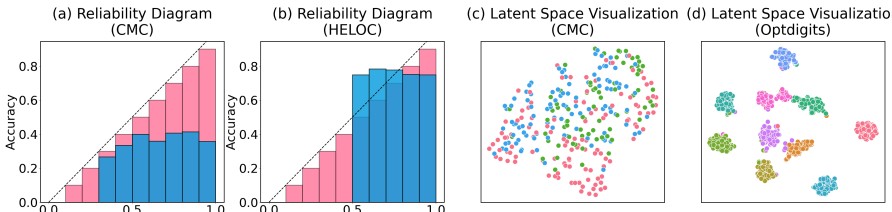

**Figure 2:** Comparison of reliability diagrams for both overconfident (a) and under-confident (b) prediction probabilities in the same tabular domain, along with the comparison of the latent spaces of tabular (c) and image data (d) using t-SNE within the same MLP model architecture, where different colors indicate different classes.

## 2 Design Principles on Test-Time Adaptation for Tabular Data

In this section, to establish the design principles of TTA on tabular data, we conduct a comprehensive analysis of the shortcomings observed in existing test-time adaptation strategies, including entropy minimization-based approaches, as outlined in Section 2.1. Furthermore, we discuss the issue of label distribution shift between source and target domains, along with addressing the class imbalance problem in Section 2.2.

### 2.1 Failure of Existing Test-Time Adaptation Methods on Tabular Data

Domain adaptation refers to methods used in adapting machine learning models to real-world data that may differ in distribution from their training data. There are two main branches: traditional supervised/unsupervised domain adaptation (Ben-David et al., 2006; Sun et al., 2017; Ganin & Lempitsky, 2015; Khurana et al., 2021), which often require both source and target data during training, and test-time adaptation (Sun et al., 2020; Gandelsman et al., 2022; Boudiaf et al., 2022; Niu et al., 2023; Zhou et al., 2023; Park et al., 2023), a novel approach that adapts the model using only unlabeled target data during the inference phase. Indeed, developing test-time adaptation strategies for tabular data is particularly promising due to their applicability; making off-the-shelf models adaptable during deployment, which may suffer from distribution shifts during testing, all the while evading privacy concerns from viewing source data in hindsight, a crucial issue in fields such as healthcare and finance.

However, we observe that direct application entropy minimization methods and their variations, most abundant forms of fully test-time adaptation in other modalities like image (Wang et al., 2021a) and speech (Kim et al., 2023), fail to show their efficacy in the tabular domain. We find a number of grounds for this. First, we demonstrate that the distributions of prediction entropy consistently exhibit a strong bias towards the high region across various tabular datasets and model architectures, including MLP (Murtagh, 1991), TabNet (Arik & Pfister, 2021), and FT-Transformer (Gorishniy et al., 2021). As shown in Figure 1 (a) and (b), such phenomenon is unique to real tabular datasets such as CMC (Bischl et al., 2021), not linearized images like Optdigits (Alpaydin & Kaynak, 1998). This aligns with the findings in the vision domain Niu et al. (2022; 2023), where samples with high entropy generate significant gradients leading to model collapse. These works Niu et al. (2022; 2023) bypass this by filtering samples with high entropies. Nevertheless, predictions from deep learning models in the tabular domain tend to exhibit an under-confident tendency, where applying this technique to the tabular domain results that discarding samples with high entropy leaving very few samples for model update (Figure 1 (c)). In terms of probability calibration, in contrast to the generally overconfident nature of computer vision models, deep learning models trained on tabular data show overall poor calibration. However, it's worth noting that the tendency towards under-confidence or overconfidence varied across datasets (Figure 2 (a) and (b)).

We also conduct a comparison between the latent space of models trained on tabular data and those trained on linearized images, as shown in Figure 2 (c) and (d). Through this analytical exploration, it is discerned that the decision boundary within the latent space of tabular data is markedly more complex; attributable to the absence of clear class separation within the latent space. This observation provides another indication of the limitations of entropy minimization, which heavily relies on the cluster assumption – lower data density results in a closer decision boundary, leading to increased uncertainty. We also find that the majority of other TTA methods, such as test-time training methods –

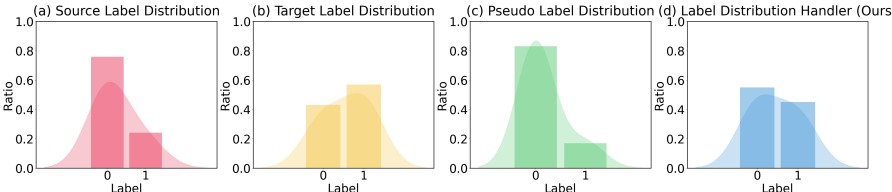

**Figure 3:** Label distribution shift between the source domain (a) and the target domain (b) in the tabular dataset – HELOC, bias of model predictions towards the source label distribution (c), and the label distribution of applying our label distribution handler towards the target label distribution (d).

TTT++ (Liu et al., 2021) and TTT-MAE (Gandelsman et al., 2022) – and output optimization methods – LAME (Boudiaf et al., 2022) and ODS (Zhou et al., 2023) – heavily rely on representation learning abilities of deep neural networks. Specifically, LAME (Boudiaf et al., 2022) and ODS (Zhou et al., 2023), directly aim to optimize the probabilities of instances proximate within the representation space, ultimately bringing them closer. However, these methods encounter limitations in tabular data because the cluster assumption does not hold. These findings underscore the need for uncertainty calibration, driven by the model's poor calibration. Conversely, they also highlight the challenges associated with improving the representation space for tabular test-time adaptation.

## 2.2 LABEL DISTRIBUTION SHIFT AND CLASS IMBALANCE PROBLEM

Label distribution shift, which is a mismatch in output class label distribution between the source domain and target domain, has been a significant challenge in various machine learning tasks (Ganin & Lempitsky, 2015; Wang et al., 2021b; Khurana et al., 2021). In situations of distribution shift in tabular data, label distribution shift commonly occurs because of factors such as data collection patterns, cohort differences, and spatial and temporal changes. These challenges can significantly impact model performance, emphasizing the importance of considering them in the context of tabular test-time adaptation.

Recently, in the field of test-time adaptation for image classification, methods have been proposed to address such issue of label distribution shifts (Park et al., 2023; Zhou et al., 2023). However, these methods make the impractical assumption that the source model is trained on perfectly class-balanced data. This assumption might hold for standard image classification benchmarks like MNIST (Deng, 2012) or CIFAR-10 (Krizhevsky et al., 2009) but is hard to be satisfied in the tabular domain. Under the class imbalanced scenario of the source domain, we find that models tend to produce biased predictions during testing based on the class distribution observed in the training data (Figure 3). These findings pose an additional challenge of addressing both label distribution shift and class imbalance, thus highlighting the need for developing a novel test-time adaptation method for the tabular domain.

Before delving into the intricacies of our approach, we highlight a pivotal finding: uncertainty calibration plays a crucial role in mitigating the label distribution shifts. Our method fundamentally estimates the average label distribution of the current batch and it corrects predictions by maintaining confident predictions while adjusting uncertain predictions towards the estimated average label distribution of the current batch. As shown in Table 1, we observe that if the source model is perfectly calibrated by increasing the confidence for correct samples while decreasing the confidence for incorrect samples, our label distribution handler leads to a remarkable improvement in performance. This highlights the critical role of developing an effective uncertainty calibrator for the label distribution handler.

**Table 1:** Pivotal findings that uncertainty calibration benefits the label distribution handler.

| Method | HELOC | ANES |
|---|---|---|
| Unadapted | 47.6 | 79.3 |
| AdapTable | 63.7 | 79.6 |
| **AdapTable (Oracle)** | **90.1** | **84.7** |

## 3 ADAPTABLE

In this section, we introduce an **Adap**tation for **Table** (*AdapTable*) framework, which is the first tabular test-time adaptation strategy. AdapTable focuses on correcting the output probability and circumvents the need to tune the model parameter using unsupervised objectives. To this end, we aim to maintain predictions for confident test samples while heavily correcting predictions for uncertain test samples. To estimate the model's uncertainty in cases of *poorly calibrated source models* (as

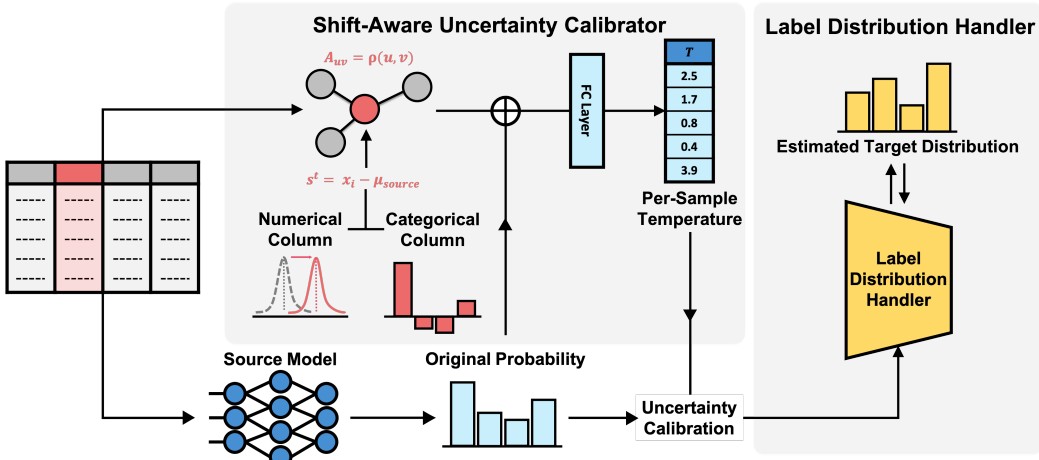

**Figure 4:** The overall pipeline of the proposed *AdapTable* framework. AdapTable focuses on correcting the output probability and circumvents the need to tune the model parameters. Towards this, AdapTable estimates the target label distribution (Section 3.3) and adjusts the original probability based on calibrated uncertainty (Section 3.2).

investigated in Section 2.1), we derive a shift-aware uncertainty calibrator, which predicts per-sample temperature scaling factor to calibrate model predictions in Section 3.2. After that, a novel label distribution handler is introduced in Section 3.3. The overall framework of the proposed method is depicted in Figure 4 and in Algorithm 1.

## 3.1 TEST-TIME ADAPTATION SETUP FOR TABULAR DATA

We first define the problem setting of test-time adaptation for the tabular domain. Let $\mathcal{F}(\cdot|\theta) : \mathbb{R}^D \to \mathbb{R}^C$ be a pre-trained classifier on the labeled source tabular domain $\mathcal{D}_s = \{(\boldsymbol{x}_i^s, y_i^s)\}_i$ in pairs of a tabular input and its output class label, which takes a row $\boldsymbol{x}_i \in \mathbb{R}^D$ from a table and returns output logit $\mathcal{F}(\boldsymbol{x}_i|\theta)$. Here, $D$ and $C$ are the number of input features and output classes, respectively. We focus on tabular classification, and suggest test-time adaptation methods for tabular classification aim to adapt $\mathcal{F}(\cdot|\theta)$ to the unlabeled target tabular domain $\mathcal{D}_t = \{\boldsymbol{x}_i^t\}_i$ during the test phase without access to $\mathcal{D}_s$. Unlike most TTA approaches which directly fine-tune $\theta$ with unsupervised objectives, we correct the output prediction $\mathcal{F}(\boldsymbol{x}_i^t|\theta)$ directly.

## 3.2 SHIFT-AWARE UNCERTAINTY CALIBRATOR

Since source tabular models are poorly calibrated as discussed in Section 2.1 and uncertainty calibration is helpful for handling label distribution shifts, given a test batch $\{\boldsymbol{x}_i^t\}_{i=1}^N$ in the target domain, we propose a *shift-aware post-hoc uncertainty calibrator* $\mathcal{G}(\cdot, \cdot|\phi) : \mathbb{R}^{NC} \times \mathbb{R}^D \to \mathbb{R}$ that incorporates shift information and simultaneously addresses both overconfidence and under-confidence, departing from traditional approaches that primarily rely on output logits only (Guo et al., 2017) and focus only on either overconfidence (Guo et al., 2017; Pollastri et al., 2021) or under-confidence (Wang et al., 2021c; Hsu et al., 2022). Here, $\mathcal{G}$ takes not only the output logit $\mathcal{F}(\boldsymbol{x}_i^t|\theta)$ of each test instance $\boldsymbol{x}_i^t$ but also the shift information of current batch $\boldsymbol{s}^t$ as additional information and returns a per-sample temperature scaling factor $T_i = \mathcal{G}(\mathcal{F}(\boldsymbol{x}_i^t|\theta), \boldsymbol{s}^t|\phi)$. We define shift information $\boldsymbol{s}_u^t$ of a specific column index $u$ as follows:

$$\boldsymbol{s}_u^t = \left(\boldsymbol{x}_{iu}^t - \Big(\sum_{i'=1}^{|\mathcal{D}_s|} \boldsymbol{x}_{i'}^s/|\mathcal{D}_s|\Big)_u\right)_{i=1}^N.$$

Here, we focus on the distribution shift of each column, which is a *tabular-specific* input covariate shift, and form a shift-aware graph, where each node represents a specific column, and each edge represents the relationship between different columns. Specifically, we form a node feature $\boldsymbol{h}_u^{(0)}$ of

each node index $u$ and adjacency matrix $\boldsymbol{A} \in [0, 1]^{D \times D}$ as follows:

$$\boldsymbol{h}_u^{(0)} = \boldsymbol{s}_u^t \quad \text{for } 1 \leq u \leq D$$
$$\boldsymbol{A}_{uv} = \rho\big((\boldsymbol{x}_{iu}^t)_{i=1}^N, (\boldsymbol{x}_{iv}^t)_{i=1}^N\big) \quad \text{for } 1 \leq u, v \leq D,$$

where $\rho(X, Y)$ denotes the absolute value of correlation coefficient between $X$ and $Y$. Such node embeddings are reasonable for the tabular domain, where each column has a distinct and independent meaning, and these embeddings consider the shifts and relationships between them. In our $\mathcal{G}$, we have columnwise projection layer $l_u$ (for matching input dimensions between numerical and categorical columns – we assume that all columns are numerical for the simplicity of notation, but in situations, where categorical columns are present, $\boldsymbol{s}_u^t$ for categorical column index $u$ becomes a matrix instead of a single vector and apply linear projection on it under the one-hot encoding scheme), single graph convolution layer GCN$(\cdot, \cdot, \cdot)$, and final decoder Dec$(\cdot)$ that returns per-sample temperature scaling factor $T_i$. After forming a shift-aware graph, we apply $l_u$ and GCN to get hidden representation $\boldsymbol{h}_u^{(1)} = \text{GCN}\big(l_u \boldsymbol{h}_u^{(0)}, \{l_v \boldsymbol{h}_v^{(0)}, \forall v \in \mathcal{N}_u\}, \boldsymbol{A}\big)$, where $\mathcal{N}_u$ denotes the index set of the neighborhoods of $u$-th column (*i.e.*, all other columns defined by $\boldsymbol{A}$). We then average $\boldsymbol{h}_u^{(1)}$ to extract a comprehensive representation $\tilde{\boldsymbol{h}} = \sum_{u=1}^D \boldsymbol{h}_u^{(1)}/D$ to capture the shift information across all columns within current test batch, and finally get per-sample temperature scaling factor $T_i = \text{Dec}\big(\mathcal{F}(\boldsymbol{x}_i^t|\theta), \tilde{\boldsymbol{h}}\big)$. We train $\mathcal{G}$ by optimizing $\phi$ using training set again after training $\mathcal{F}$ with loss unction $\mathcal{L} = \mathcal{L}_{\text{FL}} + \lambda_{\text{CAL}}\mathcal{L}_{\text{CAL}}$ motivated by under-confidence behavior in Figure 1 (a) and Figure 2 (b), where $\mathcal{L}_{\text{FL}}$ denotes focal loss (Lin et al., 2017), $\mathcal{L}_{\text{CAL}}$ denotes a regularization term suggested in CaGCN (Wang et al., 2021c), and $\lambda_{\text{CAL}}$ denotes the weight of regularization term. Such post-training strategy is commonly used in test-time adaptation methods (Lim et al., 2023; Park et al., 2023), and the detailed post-training strategy is fully described in Section C.

## 3.3 Label Distribution Handler

This section proposes a *label distribution handler*, for correctly estimating the target label distribution on the current test batch. Formally, if we estimate the class probability given $\boldsymbol{x}_i^t$ in the target domain $\boldsymbol{p}_t(y|\boldsymbol{x}_i^t) = \sigma_{\text{Softmax}}(\mathcal{F}(\boldsymbol{x}_i^t|\theta))$ with Softmax function $\sigma_{\text{Softmax}}$, using Bayes' theorem, this can be represented as follows:

$$\boldsymbol{p}_t(y|\boldsymbol{x}_i^t) = \frac{\boldsymbol{p}_t(y)\boldsymbol{p}_t(\boldsymbol{x}_i^t|y)}{\boldsymbol{p}_t(\boldsymbol{x}_i^t)}.$$

Based on the observation that the distribution of the model in the target domain is biased towards the label distribution of the source domain as discussed in Section 2.2 and Figure 3, we can infer that $\boldsymbol{p}_t(y)$ is highly biased to $\boldsymbol{p}_s(y) = \big(\sum_{i=1}^{|\mathcal{D}_s|} \mathbb{1}_{[j=y_i^s]}/|\mathcal{D}_s|\big)_{j=1}^C$ with an indicator function $\mathbb{1}$. One simple solution to correct this biased probability is to multiply $\boldsymbol{p}_t(y)/\boldsymbol{p}_s(y)$, *i.e.*, use $\boldsymbol{p}_t(y|\boldsymbol{x}_i^t) \cdot \boldsymbol{p}_t(y)/\boldsymbol{p}_s(y)$ as our final prediction. We assume that we can access the marginal label distribution of the source domain similar to TTT++ (Liu et al., 2021). To estimate the marginal label distribution of the target domain $\boldsymbol{p}_t(y)$ without labels, we derive an *online target label estimator* to leverage the temporal locality of label distribution in real-world scenarios, where class labels of test samples collected at close time intervals tend to exhibit relative similarity. Additionally, we introduce a *debiased target label estimator* to mitigate bias towards the source label distribution. At first, we initialize online target label estimator $\boldsymbol{p}_t^{\text{oe}}(y) = (1/C)_{j=1}^C$ as a uniform distribution. Given a test batch $\{\boldsymbol{x}_i^t\}_{i=1}^N$, we predict the debiased target label estimator $\boldsymbol{p}_t^{\text{de}}(y|\boldsymbol{x}_i^t)$ and estimate the current target label distribution $\boldsymbol{p}_t(y)$ as follows:

$$\boldsymbol{p}_t^{\text{de}}(y|\boldsymbol{x}_i^t) = \sigma_{L_1}\Big(\boldsymbol{p}_t(y|\boldsymbol{x}_i^t)/\boldsymbol{p}_s(y)\Big)$$

$$\boldsymbol{p}_t(y) = (1 - \alpha) \cdot \sum_{i=1}^N \boldsymbol{p}_t^{\text{de}}(y|\boldsymbol{x}_i^t)/N + \alpha \cdot \boldsymbol{p}_t^{\text{oe}}(y),$$

where the interpolate cumulative online target label estimator and the average probability of the current debiased target label estimator with $L_1$ normalization function $\sigma_{L_1}$. Now, we utilize the shift-aware uncertainty calibrator defined in Section 3.2. Given a current test batch $\{\boldsymbol{x}_i^t\}_{i=1}^N$, we calculate $\boldsymbol{s}^t$ and get per-sample temperature $T_i = \mathcal{G}(\mathcal{F}(\boldsymbol{x}_i^t|\theta), \boldsymbol{s}^t|\phi)$. We define the uncertainty $\epsilon_i$

**Table 2:** The average accuracy (%) and their corresponding standard errors for both supervised models and TTA baselines are reported across three representative backbone architectures and three datasets from TableShift benchmark, which include natural distribution shifts. The results are reported over three different random seeds.

| | Method | HELOC | | | ANES | | | DIABETES READMISSION | | |
| | | Acc. | bAcc. | F1 | Acc. | bAcc. | F1 | Acc. | bAcc. | F1 |
|---|---|---|---|---|---|---|---|---|---|---|
| **Supervised** | K-NN | 47.7 ±0.0 | 62.0 ±0.0 | 40.3 ±0.0 | 74.8 ±0.0 | 76.9 ±0.0 | 71.1 ±0.0 | 57.4 ±0.0 | 57.7 ±0.0 | 56.9 ±0.0 |
| | LR | 49.9 ±0.0 | 63.5 ±0.0 | 44.2 ±0.0 | 78.7 ±0.0 | 80.2 ±0.0 | 76.2 ±0.0 | 60.2 ±0.0 | 61.4 ±0.0 | 58.9 ±0.0 |
| | RF | 44.1 ±0.5 | 58.2 ±7.6 | 32.2 ±1.1 | 74.3 ±0.3 | **81.7** ±0.1 | 68.4 ±0.6 | 53.7 ±0.4 | **64.4** ±0.4 | 42.1 ±1.2 |
| | XGBOOST | 48.0 ±2.5 | 57.6 ±7.2 | 39.9 ±4.9 | 78.5 ±0.2 | 80.5 ±0.2 | 75.8 ±0.3 | 62.2 ±0.1 | 63.1 ±0.1 | 61.3 ±0.2 |
| | CATBOOST | 54.7 ±0.0 | 65.4 ±0.0 | 51.7 ±0.0 | 79.1 ±0.0 | 80.4 ±0.0 | 76.8 ±0.0 | 62.6 ±0.2 | 63.4 ±0.0 | 61.8 ±0.3 |
| | **+ ADAPTABLE** | 65.6 ±0.0 | 65.5 ±0.0 | 65.4 ±0.0 | 79.9 ±0.0 | 79.6 ±0.0 | 78.6 ±0.0 | 62.9 ±0.1 | 63.3 ±0.0 | 62.5 ±0.1 |
| **MLP** | Unadapted | 47.0 ±1.6 | 53.2 ±1.3 | 38.2 ±3.2 | 79.3 ±0.2 | 76.5 ±0.4 | 77.3 ±0.3 | 61.3 ±0.1 | 61.1 ±0.1 | 60.2 ±0.1 |
| | PL | 45.3 ±1.0 | 51.8 ±0.8 | 34.9 ±2.1 | 78.9 ±0.2 | 75.6 ±0.4 | 76.6 ±0.4 | 60.7 ±0.1 | 60.5 ±0.1 | 58.9 ±0.1 |
| | TTT++ | 47.0 ±1.6 | 53.2 ±1.3 | 38.2 ±3.2 | 79.5 ±0.1 | 76.8 ±0.3 | 77.6 ±0.2 | 61.3 ±0.1 | 61.1 ±0.1 | 60.2 ±0.1 |
| | TENT | 44.6 ±1.2 | 51.2 ±1.0 | 33.2 ±2.4 | 78.0 ±0.3 | 74.0 ±0.5 | 74.9 ±0.5 | 60.2 ±0.2 | 60.2 ±0.1 | 58.3 ±0.1 |
| | SAM | 43.1 ±0.0 | 50.0 ±0.0 | 30.1 ±0.0 | 70.8 ±0.8 | 63.8 ±1.1 | 62.0 ±1.6 | 59.6 ±0.8 | 59.3 ±0.9 | 56.5 ±2.0 |
| | EATA | 47.0 ±1.6 | 53.2 ±1.3 | 38.2 ±3.2 | 79.3 ±0.2 | 76.5 ±0.4 | 77.3 ±0.3 | 61.3 ±0.1 | 61.1 ±0.1 | 60.2 ±0.1 |
| | SAR | 43.1 ±0.0 | 50.0 ±0.0 | 30.1 ±0.0 | 69.4 ±0.7 | 62.0 ±0.9 | 59.4 ±1.4 | 57.5 ±0.8 | 57.1 ±0.9 | 51.3 ±2.1 |
| | LAME | 43.1 ±0.0 | 50.0 ±0.0 | 30.1 ±0.0 | 63.5 ±0.3 | 54.6 ±0.4 | 46.8 ±0.7 | 55.3 ±0.4 | 54.9 ±0.4 | 46.9 ±0.8 |
| | **ADAPTABLE** | **64.5** ±0.3 | **65.8** ±0.4 | **64.5** ±0.3 | 79.6 ±0.1 | **78.4** ±0.2 | 78.6 ±0.0 | 61.7 ±0.0 | **61.7** ±0.0 | 61.7 ±0.0 |
| **TabNet** | Unadapted | 43.1 ±0.0 | 50.0 ±0.0 | 30.2 ±0.1 | 72.0 ±3.3 | 67.0 ±3.9 | 66.8 ±4.7 | 53.9 ±3.3 | 53.4 ±3.4 | 42.2 ±8.6 |
| | PL | 43.1 ±0.0 | 50.0 ±0.0 | 30.2 ±0.1 | 71.6 ±3.2 | 66.4 ±3.8 | 66.0 ±4.5 | 53.9 ±3.3 | **53.5** ±3.4 | 42.8 ±8.4 |
| | TTT++ | 43.1 ±0.0 | 50.0 ±0.0 | 30.2 ±0.1 | 71.6 ±3.4 | 65.9 ±4.9 | 64.6 ±6.7 | 53.0 ±2.3 | 52.5 ±2.3 | 40.3 ±5.9 |
| | TENT | 43.1 ±0.0 | 50.0 ±0.0 | 30.2 ±0.1 | 73.7 ±1.5 | 68.5 ±2.1 | 68.6 ±2.5 | 53.1 ±2.2 | 52.5 ±2.3 | 41.4 ±5.6 |
| | SAM | 43.1 ±0.0 | 50.0 ±0.0 | 30.2 ±0.1 | 69.7 ±3.5 | 64.8 ±4.9 | 63.9 ±5.6 | 50.6 ±0.0 | 50.0 ±0.0 | 33.6 ±0.0 |
| | EATA | 43.1 ±0.0 | 50.0 ±0.0 | 30.2 ±0.1 | 74.4 ±1.3 | 69.4 ±1.9 | 69.7 ±2.2 | 53.0 ±2.3 | 52.5 ±2.3 | 40.3 ±6.0 |
| | SAR | 43.1 ±0.0 | 50.0 ±0.0 | 30.1 ±0.0 | 72.5 ±2.3 | 68.5 ±3.6 | 68.3 ±4.1 | 50.6 ±0.1 | 50.0 ±0.1 | 34.4 ±0.7 |
| | LAME | 43.1 ±0.0 | 50.0 ±0.0 | 30.1 ±0.0 | 67.8 ±4.6 | 60.8 ±6.6 | 55.6 ±10.4 | 50.6 ±0.0 | 50.0 ±0.0 | 33.9 ±0.3 |
| | **ADAPTABLE** | **51.5** ±4.6 | **55.6** ±3.0 | **46.1** ±6.9 | **75.2** ±1.1 | **73.0** ±2.1 | **72.8** ±2.1 | **56.1** ±3.0 | 50.2 ±0.0 | **55.8** ±2.8 |
| **FT-Transformer** | Unadapted | 43.4 ±0.3 | 50.3 ±0.2 | 30.9 ±0.8 | 79.1 ±0.1 | 75.9 ±0.4 | 76.8 ±0.3 | 62.5 ±0.1 | 62.4 ±0.1 | 61.9 ±0.2 |
| | PL | 43.2 ±0.1 | 50.0 ±0.0 | 30.3 ±0.2 | 78.8 ±0.2 | 75.3 ±0.5 | 76.3 ±0.5 | 62.2 ±0.1 | 62.0 ±0.1 | 61.3 ±0.3 |
| | TTT++ | 43.5 ±0.4 | 50.3 ±0.3 | 31.0 ±0.9 | **79.3** ±0.1 | 76.6 ±0.3 | 77.4 ±0.2 | **62.6** ±0.0 | **62.5** ±0.0 | **62.2** ±0.1 |
| | TENT | 43.2 ±0.1 | 50.1 ±0.1 | 30.4 ±0.2 | 78.5 ±0.3 | 74.9 ±0.6 | 75.9 ±0.6 | 62.2 ±0.2 | 62.0 ±0.2 | 61.2 ±0.3 |
| | SAM | 43.1 ±0.0 | 50.0 ±0.0 | 30.1 ±0.0 | 73.8 ±1.2 | 67.8 ±1.6 | 67.5 ±2.0 | 59.4 ±0.5 | 59.0 ±0.5 | 55.2 ±1.0 |
| | EATA | 47.9 ±4.3 | 53.4 ±3.1 | 39.4 ±8.1 | 79.1 ±0.1 | 75.9 ±0.4 | 76.8 ±0.3 | 62.5 ±0.1 | 62.4 ±0.1 | 61.9 ±0.2 |
| | SAR | 43.3 ±0.2 | 50.2 ±0.2 | 30.6 ±0.5 | 74.2 ±1.3 | 68.4 ±1.8 | 68.2 ±2.2 | 60.2 ±0.6 | 59.8 ±0.6 | 57.0 ±1.1 |
| | LAME | 47.7 ±4.6 | 53.3 ±3.3 | 38.9 ±8.7 | 77.2 ±0.5 | 72.9 ±0.8 | 73.7 ±0.8 | 52.1 ±0.5 | 51.6 ±0.5 | 41.3 ±2.0 |
| | **ADAPTABLE** | **60.3** ±0.5 | **60.3** ±0.5 | **55.9** ±1.5 | 77.8 ±0.2 | **77.8** ±0.2 | 78.2 ±0.0 | 61.7 ±0.0 | 60.6 ±1.8 | 60.5 ±1.8 |

of $\mathcal{F}(\boldsymbol{x}_i^t|\theta)$ as a reciprocal of margin of calibrated probability distribution $\sigma_{\text{Softmax}}(\mathcal{F}(\boldsymbol{x}_i^t|\theta)/T_i)$ (Section C). We then measure quantiles of each instance $\boldsymbol{x}_i$ using $\epsilon_i$ within current batch, and adjust original probability with $c_i$ as $\boldsymbol{p}_t(y|\boldsymbol{x}_i^t)' = \sigma_{\text{Softmax}}(c_i \cdot \mathcal{F}(\boldsymbol{x}_i^t|\theta))$ with a scaling constant $c_i$ defined like below:

$$c_i = \begin{cases} T & \text{if} \quad \epsilon_i \leq Q\big(\{\epsilon_{i'}\}_{i'=1}^N, q_{\text{low}}\big) \\ 1 & \text{if} \quad Q\big(\{\epsilon_{i'}\}_{i'=1}^N, q_{\text{low}}\big) < \epsilon_i < Q\big(\{\epsilon_{i'}\}_{i'=1}^N, q_{\text{high}}\big) \\ 1/T & \text{if} \quad \epsilon_i \geq Q\big(\{\epsilon_{i'}\}_{i'=1}^N, q_{\text{high}}\big), \end{cases} \tag{1}$$

where $Q(X, q)$ denotes the quantile function which gives the value corresponding to the lower $q$ quantile in $X$, $T = 3 \max_j \boldsymbol{p}_s(y)_j / 2 \min_j \boldsymbol{p}_s(y)_j$ denotes temperature scaling hyperparameter, and $q_{\text{low}}/q_{\text{high}}$ denote low/high quantiles, respectively. Finally, we get the final prediction $\hat{\boldsymbol{p}}_i(y)$ with self-ensembling (Gao et al., 2023) and updates online target label estimator $\boldsymbol{p}_t^{\text{oe}}(y)$ as follows:

$$\hat{\boldsymbol{p}}_i(y) = \boldsymbol{p}_t(y|\boldsymbol{x}_i^t)'/2 + \sigma_{L_1}\Big(\boldsymbol{p}_t(y|\boldsymbol{x}_i^t)'\boldsymbol{p}_t(y)/\boldsymbol{p}_s(y)\Big)/2$$

$$\boldsymbol{p}_t^{\text{oe}}(y) \leftarrow (1-\alpha) \cdot \sum_{i=1}^N \hat{\boldsymbol{p}}_i(y)/N + \alpha \cdot \boldsymbol{p}_t^{\text{oe}}(y).$$

The more detailed explanation of AdapTable and the algorithm table can be found in Section C and Algorithm 1, respectively.

## 4 EXPERIMENTS

### 4.1 EXPERIMENTAL SETUP

**Source Tabular Models and Datasets** To verify the proposed method under various tabular models, three representative deep tabular learning models – MLP (Murtagh, 1991), TabNet (Arik

**Table 3:** The average accuracy (%) and their corresponding standard errors of both supervised models and TTA baselines with MLP backbone architecture across three datasets from OpenML-CC18 benchmark (Bischl et al., 2021), each corrupted with 6 different synthetic corruptions: Gaussian noise, uniform noise, random column drop, random missing, numerical column shifts, categorical column shifts. The results are reported over three different random seeds.

| | Method | CMC | | | MFEAT-PIXEL | | | DNA | | |
|---|---|---|---|---|---|---|---|---|---|---|
| | | Acc. | bAcc. | F1 | Acc. | bAcc. | F1 | Acc. | bAcc. | F1 |
| Supervised | K-NN | 47.9 ±1.4 | 45.4 ±1.1 | 44.0 ±1.1 | 96.2 ±0.3 | 96.1 ±0.3 | 95.8 ±0.4 | 79.3 ±1.3 | 77.0 ±0.9 | 77.8 ±1.0 |
| | LR | 49.0 ±1.2 | 46.3 ±1.1 | 42.8 ±1.1 | 96.8 ±0.3 | 96.6 ±0.3 | 96.6 ±0.3 | 89.4 ±1.6 | 87.9 ±1.5 | 87.8 ±1.7 |
| | RF | 51.4 ±1.3 | 48.5 ±1.1 | 44.5 ±1.2 | 96.5 ±0.2 | 96.3 ±0.2 | 96.3 ±0.2 | 91.1 ±1.7 | 90.7 ±1.4 | 89.2 ±2.0 |
| | XGBOOST | 52.9 ±1.6 | 49.7 ±1.4 | 47.1 ±1.5 | 94.2 ±0.6 | 94.0 ±0.6 | 93.9 ±0.6 | 91.8 ±1.7 | 90.7 ±1.4 | 89.2 ±2.0 |
| | CATBOOST | 51.9 ±1.6 | 49.0 ±1.5 | 47.4 ±1.6 | 96.5 ±0.3 | 96.3 ±0.3 | 96.3 ±0.3 | 91.8 ±1.7 | 90.7 ±1.6 | 90.4 ±1.9 |
| | + ADAPTABLE | 52.4 ±0.7 | 49.3 ±0.5 | 48.6 ±0.6 | 96.7 ±0.3 | 96.6 ±0.3 | 96.5 ±0.3 | 94.6 ±0.7 | 92.9 ±1.6 | 93.8 ±1.3 |
| MLP | Unadapted | 53.5 ±1.4 | 47.9 ±1.2 | 47.9 ±1.2 | 96.4 ±0.2 | 96.4 ±0.2 | 96.3 ±0.2 | 91.4 ±1.2 | 90.1 ±1.4 | 90.0 ±1.3 |
| | PL | 54.1 ±1.6 | 47.8 ±1.3 | 47.8 ±1.3 | 96.4 ±0.2 | 96.5 ±0.2 | 96.3 ±0.2 | 90.9 ±1.3 | 89.5 ±1.6 | 89.4 ±1.4 |
| | TTT++ | 52.7 ±1.4 | 47.2 ±1.2 | 47.2 ±1.2 | 95.8 ±0.2 | 95.6 ±0.3 | 95.5 ±0.3 | 89.5 ±1.1 | 86.6 ±1.3 | 87.6 ±1.1 |
| | TENT | 54.1 ±1.6 | 47.5 ±1.2 | 47.5 ±1.2 | 96.5 ±0.2 | 96.5 ±0.2 | 96.4 ±0.2 | 90.3 ±1.3 | 88.7 ±1.7 | 88.6 ±1.5 |
| | SAM | 49.5 ±1.7 | 39.3 ±0.6 | 39.3 ±0.6 | 66.9 ±3.4 | 65.2 ±3.5 | 64.3 ±3.8 | 67.0 ±1.7 | 60.8 ±3.0 | 58.1 ±2.7 |
| | EATA | 53.8 ±1.4 | 48.1 ±1.2 | 48.1 ±1.2 | 96.4 ±0.2 | 96.5 ±0.2 | 96.3 ±0.2 | 91.4 ±1.2 | 90.1 ±1.5 | 90.0 ±1.3 |
| | SAR | 48.0 ±1.7 | 37.7 ±0.6 | 37.7 ±0.6 | 65.6 ±3.8 | 63.8 ±3.9 | 62.8 ±4.1 | 62.6 ±2.6 | 59.8 ±2.6 | 55.2 ±3.2 |
| | LAME | 49.6 ±3.7 | 40.3 ±2.7 | 40.3 ±2.7 | 95.9 ±0.5 | 95.8 ±0.5 | 95.7 ±0.5 | 74.9 ±3.8 | 63.6 ±5.1 | 63.8 ±6.4 |
| | ADAPTABLE | 55.7 ±2.0 | 50.3 ±1.5 | 50.2 ±1.5 | 97.8 ±0.2 | 97.5 ±0.2 | 97.4 ±0.2 | 95.0 ±0.5 | 89.9 ±1.6 | 92.1 ±0.7 |

& Pfister, 2021), and FT-Transformer (Gorishniy et al., 2021) – are used as source tabular models. To demonstrate our approach across various test-time distribution shifts, we test our method under three natural distribution shifts datasets – HELOC (Brown et al., 2018), ANES (Studies, 2022), and DIABETES READMISSION (Clore et al., 2014) in TableShift benchmark (Gardner et al., 2023). Furthermore, we also verify our method by randomly splitting three datasets – CMC, MFEAT-PIXEL, and DNA – within the OpenML-CC18 benchmark (Bischl et al., 2021), and injecting six types of synthetic corruptions – Gaussian noise, uniform noise, random missing, column drop, numerical column shift, and categorical column shift – on them. These datasets are selected with the following careful considerations: i) consist of both numerical and categorical features, ii) consist only of categorical or numerical features, and iii) different task types including both binary and multi-way classification. More details about the datasets are described in Section E.1 of the supplementary material.

**Baselines and Implementation Details**  We compare AdapTable with seven test time adaptation baselines – PL (Lee, 2013), TTT++ (Liu et al., 2021), TENT (Wang et al., 2021a), SAM (Foret et al., 2021) (with TENT), EATA (Niu et al., 2022), SAR (Niu et al., 2023), and LAME (Boudiaf et al., 2022) – in the other domain. For the purpose of performance reference, we also report the accuracy of classical machine learning algorithms: k-nearest neighbors (K-NN), logistic regression (LR), random forest (RF), XGBOOST (Chen & Guestrin, 2016), and CATBOOST (Dorogush et al., 2017). We employ fixed batch size 64 for all experiments, a common setting among baselines (Schneider et al., 2020; Wang et al., 2021a). Other test-time hyperparameters are tuned with respect to numerical column shift within the CMC dataset, for each backbone architecture. The hyperparameters are then fixed throughout all of the datasets. More details about the baselines and implementations are shown in Section F.2 and Section F.3 of the supplementary material.

## 4.2  MAIN RESULTS

**Result on Natural Distribution Shifts**  Table 2 shows the result on natural distribution shifts. We observe that our method consistently outperforms baselines in most settings across different datasets and backbone architecture types. Furthermore, it is worth noting that our method, integrated with CatBoost (Dorogush et al., 2017), can also effectively adapt in these settings, where baseline TTA approaches cannot be applicable. As discussed in 2.1, entropy minimization variants fail under most datasets, as well as the baselines that depend on cluster assumptions on latent space - TTT++ (Liu et al., 2021), EATA (Niu et al., 2022) and LAME (Boudiaf et al., 2022).

**Result on Synthetic Corruptions**  We further evaluate the efficacy of AdapTable on different datasets of OpenML-CC18 benchmark (Bischl et al., 2021) on synthetic corruptions, depicted in Table 3. Please note that we report the average performance across six synthetic corruptions. The overall trend is similar to real-world distributional shifts – showing state-of-the-art performance on

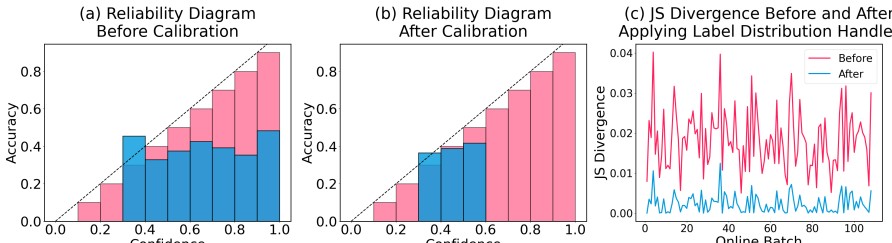

**Figure 5:** Reliability diagram before (a) and after (b) applying shift-aware uncertainty calibrator for CMC dataset, and Jensen-Shannon Divergence between estimated target label distribution before and after applying label distribution handler upon HELOC dataset (c).

most datasets and evaluation metrics, showing that our method can effectively adapt under various datasets with both numerical and categorical columns (CMC), with only numerical (MFEAT-PIXEL) and categorical columns (DNA). It is also worth highlighting that our approach can be combined with CatBoost in this setting as well for these synthetic corruptions.

### 4.3 FURTHER ANALYSIS

**Ablation Study**   To prove each component of the proposed method towards performance improvement, we conduct an ablation study. As shown in Table 4, we believe that the proposed shift-aware uncertainty calibrator as well as the graph convolution layer within it effectively increases the accuracy under both natural distribution shifts (HELOC) and synthetic corruptions (MFEAT-PIXEL, CMC).

**Table 4:** Ablation study of AdapTable decomposed into the components, where *SUC* indicates shift-aware uncertainty calibrator.

| METHOD | CMC | | MFEAT-PIXEL | HELOC |
| | NUMERICAL | CATEGORICAL | | |
| --- | --- | --- | --- | --- |
| Unadapted | $59.7 \pm 1.6$ | $58.1 \pm 0.8$ | $96.0 \pm 0.1$ | $47.0 \pm 1.6$ |
| AdapTable *(w/o SUC)* | $61.9 \pm 0.7$ | $61.4 \pm 1.2$ | $97.2 \pm 0.1$ | $61.7 \pm 2.2$ |
| AdapTable *(w/ MLP SUC)* | $64.0 \pm 0.9$ | $62.6 \pm 1.3$ | $97.4 \pm 0.1$ | $62.5 \pm 0.6$ |
| **AdapTable** | $\mathbf{65.6 \pm 1.6}$ | $\mathbf{64.2 \pm 2.4}$ | $\mathbf{97.8 \pm 0.2}$ | $\mathbf{64.5 \pm 0.3}$ |

**Efficacy of Shift-Aware Uncertainty Calibrator**   To verify the proposed shift-aware uncertainty calibrator suggested in Section 3.3, we compare the reliability diagram before and after applying shift-aware uncertainty calibrator in Figure 5 (a) and (b). The proposed method's ability not only to bring the reliability diagram closer to the $y = x$ line but also to reduce confidence and increase uncertainty for shifted test sets is indeed noteworthy.

**Efficacy of Label Distribution Handler**   Figure 5 (c) compares Jensen-Shannon Divergence value of test label distribution and prediction on each batch, between before and after adaptation using label distribution handler (LDH) for HELOC (Brown et al., 2018) dataset. The figure represents a large amount of decline of divergence, which implies label distribution handler 3.3 can indeed reduce a gap between shifted target label distribution and source-oriented model prediction.

## 5 CONCLUSION

In this paper, we have introduced *AdapTable*, a tabular-specific test-time adaptation strategy for the first time. We have systematically investigated the challenges related to test-time adaptation for tabular data and have proposed test-time adaptation strategies to overcome such problems with two key components: a shift-aware uncertainty calibrator, which utilizes information about covariate shifts among columns to correct poor confidence calibration in the model, and a label distribution handler, which estimates the label distribution of the current test batch in real-time and uses the improved uncertainty to adjust the output distribution. Extensive experiments have demonstrated that the proposed method achieves state-of-the-art performance across different datasets with both natural distribution shifts and synthetic corruptions and three representative deep tabular learning architectures. AdapTable provides a foundational philosophy for designing test-time adaptation in the context of tabular data and sheds light on the domain adaptation field for tabular data, offering valuable insights and directions for future research.

ETHICS STATEMENT

Tabular data often contains sensitive personal information, such as medical test results in healthcare data or financial details in economic data. Therefore, it is crucial to handle such data with utmost care and consideration for privacy and security concerns. In this regard, *AdapTable*, which enables the adaptation of a model to the target domain without requiring access to sensitive source domain data, holds significant appeal. We believe that the proposed method has the potential to address privacy and security concerns associated with sensitive source domain data while still achieving effective adaptation to the target domain.

REPRODUCIBILITY STATEMENT

To ensure reproducibility, we will release the source code soon.

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

# APPENDIX

## A   RELATED WORKS

**Deep Tabular Learning**    The tabular domain, distinct from image and language data, has seen limited exploration in deep learning, primarily due to the absence of spatial and semantic relationships that convolutional or recurrent neural networks rely on. Although attention-based architectures like TabNet (Arik & Pfister, 2021), FT-Transformer (Gorishniy et al., 2021) have been introduced for tabular data, they often require large datasets, resulting in computational overhead that surpasses performance gains. As a result, multi-layer perceptrons (MLPs) remain dominant in practical applications.

**Test-Time Adaptation**    In recent studies, a novel approach called test-time adaptation (TTA) has emerged as a solution to address limitations found in traditional unsupervised domain adapta-

tion (UDA) methods (Ganin & Lempitsky, 2015; Wang et al., 2021b). Test-time adaptation strategies (Wang et al., 2021a; Liu et al., 2021; Niu et al., 2023; Zhou et al., 2023), aim to adapt the pre-trained model from a source domain to a target domain, without access to source data. While these TTA methods have shown promising performance, the test time adaptation scheme for the tabular domain still remains under-explored regime since the tabular domain poses unique challenges due to its own nature. Therefore, there is a strong motivation to explore and develop TTA methods tailored to the tabular realm.

**Uncertainty Calibration**   Uncertainty calibration is a crucial technique for enhancing the reliability of deep learning model outputs. It involves estimating the model's confidence in its predictions by examining the probabilities assigned to predicted classes. Conventional training methods often result in unwarranted overconfidence in model outputs, prompting research into uncertainty calibration methods, such as Platt scaling (Platt, 2000) and isotonic regression (Stylianou & Flournoy, 2002) predate the era of deep learning, while more recent methods, including beta calibration (Kull et al., 2017) and Dirichlet calibration (Kull et al., 2019) rooted in probability distributions, have emerged. Temperature scaling, akin to Platt scaling but simpler, directly impacts uncertainty while keeping model predictions intact.

**Label Distribution Shift**   In the domain adaptation area, the assumption that there is no label distribution shift can be risky, as such shifts can occur readily and significantly affect model performance (Quinonero-Candela et al., 2008). Various approaches have been developed to tackle this issue including ReMixMatch (Berthelot et al., 2020), online label adaptation (Wu et al., 2021), black box shift estimation (BBSE) (Lipton et al., 2018). Recently, in the computer vision domain, some works (Park et al., 2023; Zhou et al., 2023) have been proposed to consider label distribution shift under the test-time adaptation setting. In our work, we leverage the model's predictions and training set statistics to adjust the output. Additionally, we integrate a label adapter to effectuate substantive modifications to the output.

## B   ADDITIONAL OBSERVATIONS

### B.1   ENTROPY DISTRIBUTIONS

To show the generalizability of the observation in Section 2.1, wherein prediction entropy of the model consistently exhibits a strong bias toward the under-confident region, we additionally provide entropy distribution histograms for test instances across six different datasets and three representative deep tabular learning architectures. Here, we can observe distinct patterns between the upper four rows (HELOC, ANES, DIABETES READMISSION, CMC) and the lower two rows (MFEAT-PIXEL, DNA). In the upper four rows, the entropy is consistently high, indicating a skew towards the under-confident region. However, in the lower two rows, this is not the case. This discrepancy arises from the fact that the two datasets below are characterized by homogeneity in all columns, one being a linearized image dataset (MFEAT-PIXEL) and the other a DNA string sequence dataset (DNA). This comprehensive analysis showcases the unique characteristics of tabular data – biased entropy distributions toward the under-confident region – compared to other domains. As discussed in Section 2.1, applying entropy minimization with samples of high entropy often cause gradient exploding and model collapse (Niu et al., 2023).

### B.2   LATENT SPACE VISUALIZATIONS

We further visualize latent spaces for test instances using t-SNE across six different datasets and three representative deep tabular learning architectures, to show the common behavior of the observation in Section 2.1, wherein tabular data exhibits extremely complex decision boundary within the latent space compared to that of other domains. Again, by comparing upper for rows with tabular dataset and lower two rows with linearized image dataset (MFEAT-PIXEL) and homogeneous DNA string sequence dataset (DNA), it is obvious that the decision boundary within the latent space of tabular domain is particularly complex compared to other domains except for the case of DNA dataset with TabNet model. As shown in Section 2.1, this also provides another indication of the limitations of existing TTA methods (Sun et al., 2020; Gandelsman et al., 2022; Liu et al., 2021; Gandelsman et al., 2022; Boudiaf et al., 2022; Zhou et al., 2023), which heavily rely on the cluster assumption.

---

**Algorithm 1** AdapTable

---

1: **Input:** Pre-trained tabular classifier on the source domain $\mathcal{F}(\cdot|\theta) : \mathbb{R}^D \to \mathbb{R}^C$, post-trained shift-aware uncertainty calibrator $\mathcal{G}(\cdot, \cdot|\phi)$, indicator function $\mathbb{1}$, quantile function $Q$, Softmax and $L_1$ normalization functions $\sigma_{\text{Softmax}}(\cdot)$, $\sigma_{L_1}(\cdot)$, tabular data in the source domain $\mathcal{D}_s = \{(\boldsymbol{x}_i^s, y_i^s)\}_i$, current tabular batch in the target domain $\{\boldsymbol{x}_i^t\}_{i=1}^N$

2: **Parameters:** Smoothing factor $\alpha$, Low/high quantiles $q_{\text{low}}/q_{\text{high}}$

3: $\boldsymbol{p}_s(y), \ T \leftarrow \big(\sum_{i=1}^{|\mathcal{D}_s|} \mathbb{1}_{[j=y_i^s]}/|\mathcal{D}_s|\big)_{j=1}^C, \ 3\max_j \boldsymbol{p}_s(y)_j/2\min_j \boldsymbol{p}_s(y)_j$

4: **if** $\{\boldsymbol{x}_i^t\}_{i=1}^N$ is the first test batch **then**

5:      $\boldsymbol{p}_t^{\text{oe}}(y) \leftarrow (1/C)_{j=1}^C$                      ▷ Initialize online target label estimator

6: **end if**

7: **for** $u = 1$ to $D$ **do**

8:      $\boldsymbol{s}_u^t \leftarrow \big(\boldsymbol{x}_{iu}^t - (\sum_{i'=1}^{|\mathcal{D}_s|} \boldsymbol{x}_{i'}^s/|\mathcal{D}_s|)_u\big)_{i=1}^N$        ▷ Calculate shift information of $u$-th column

9: **end for**

10: **for** $i = 1$ to $N$ **do**

11:      $\boldsymbol{p}_t(y|\boldsymbol{x}_i^t) \leftarrow \sigma_{\text{Softmax}}\big(\mathcal{F}(\boldsymbol{x}_i^t|\theta)\big)$

12:      $\boldsymbol{p}_t^{\text{de}}(y|\boldsymbol{x}_i^t) \leftarrow \sigma_{L_1}\big(\boldsymbol{p}_t(y|\boldsymbol{x}_i^t)/\boldsymbol{p}_s(y)\big)$          ▷ Predict debiased target label estimator

13:      $T_i \leftarrow \mathcal{G}\big(\mathcal{F}(\boldsymbol{x}_i^t|\theta), \boldsymbol{s}^t|\phi\big)$            ▷ Determine per-sample temperature of $\boldsymbol{x}_i^t$

14:      $j^*, \ j^{**} \leftarrow \arg\max_{1 \le j \le C} \boldsymbol{p}_t(y|\boldsymbol{x}_i^t)_j, \ \arg\max_{1 \le j \le C, j \ne j^*} \boldsymbol{p}_t(y|\boldsymbol{x}_i^t)_j$

15:      $\epsilon_i \leftarrow 1/\big(\sigma_{\text{Softmax}}\big(\mathcal{F}(\boldsymbol{x}_i^t|\theta)/T_i\big)_{j^*} - \sigma_{\text{Softmax}}\big(\mathcal{F}(\boldsymbol{x}_i^t|\theta)/T_i\big)_{j^{**}}\big)$    ▷ Define uncertainty of $\boldsymbol{x}_i^t$

16: **end for**

17: $\boldsymbol{p}_t(y) = (1 - \alpha) \cdot \sum_{i=1}^N \boldsymbol{p}_t^{\text{de}}(y|\boldsymbol{x}_i^t)/N + \alpha \cdot \boldsymbol{p}_t^{\text{oe}}(y)$    ▷ Estimate current target label distribution

18: **for** $i = 1$ to $N$ **do**

19:      **if** $\epsilon_i \le Q\big(\{\epsilon_{i'}\}_{i'=1}^N, q_{\text{low}}\big)$ **then**

20:          $c_i \leftarrow T$

21:      **else if** $Q\big(\{\epsilon_{i'}\}_{i'=1}^N, q_{\text{low}}\big) < \epsilon_i < Q\big(\{\epsilon_{i'}\}_{i'=1}^N, q_{\text{high}}\big)$ **then**

22:          $c_i \leftarrow 1$                          ▷ Measure temperature $c_i$ using uncertainty $\epsilon_i$

23:      **else**

24:          $c_i \leftarrow 1/T$

25:      **end if**

26:      $\boldsymbol{p}_t(y|\boldsymbol{x}_i^t)' \leftarrow \sigma_{\text{Softmax}}\big(c_i \cdot \mathcal{F}(\boldsymbol{x}_i^t|\theta)\big)$           ▷ Adjust original probability with $c_i$

27:      $\hat{\boldsymbol{p}}_i(y) \leftarrow \boldsymbol{p}_t(y|\boldsymbol{x}_i^t)'/2 + \sigma_{L_1}\big(\boldsymbol{p}_t(y|\boldsymbol{x}_i^t)'\boldsymbol{p}_t(y)/\boldsymbol{p}_s(y)\big)/2$       ▷ Perform self-ensembling

28: **end for**

29: $\boldsymbol{p}_t^{\text{oe}}(y) \leftarrow (1 - \alpha) \cdot \sum_{i=1}^N \hat{\boldsymbol{p}}_i(y)/N + \alpha \cdot \boldsymbol{p}_t^{\text{oe}}(y)$      ▷ Update online target label estimator

30: **Output:** Final predictions of $\{\hat{\boldsymbol{p}}_i(y)\}_{i=1}^N$

---

### B.3 RELIABILITY DIAGRAMS

We also provide further reliability diagrams across six different datasets and three representative deep tabular learning architectures, to show that tabular data often exhibits both overconfident and under-confident patterns compared to consistent overconfident behavior in image domain (Stylianou & Flournoy, 2002), and under-confident behavior in graph domain (Wang et al., 2021c). In cases of overconfident confidence is evident, whereas in HELOC, (CMC, TabNet), and (DNA, TabNet), under-confident confidence is observed. This underscores the necessity for the proposition of uncertainty calibrators tailored specifically to the tabular domain.

## C DETAILED ALGORITHM OF ADAPTABLE

After post-training $\mathcal{G}$, we calculate shift information of $u$-th column for $1 \le u \le D$ as $\boldsymbol{s}_u^t = \big(\boldsymbol{x}_{iu}^t - (\sum_{i'=1}^{|\mathcal{D}_s|} \boldsymbol{x}_{i'}^s/|\mathcal{D}_s|)_u\big)_{i=1}^N$. Then, we predict per-sample temperature of $\boldsymbol{x}_i^t$ as $T_i = \mathcal{G}\big(\mathcal{F}(\boldsymbol{x}_i^t|\theta), \boldsymbol{s}^t|\phi\big)$ and define uncertainty $\epsilon_i$ of $\boldsymbol{x}_i^t$ as the reciprocal of the margin of the calibrated probability like below:

$$\epsilon_i = 1/\big(\sigma_{\text{Softmax}}\big(\mathcal{F}(\boldsymbol{x}_i^t|\theta)/T_i\big)_{j^*} - \sigma_{\text{Softmax}}\big(\mathcal{F}(\boldsymbol{x}_i^t|\theta)/T_i\big)_{j^{**}}\big).$$

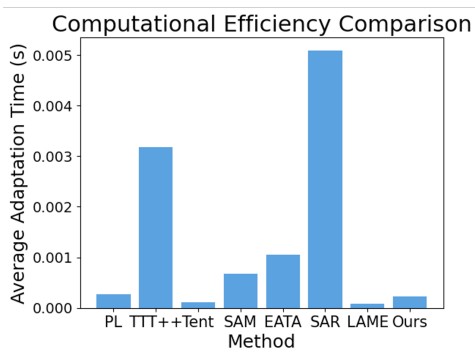

**Figure 6:** Computational efficiency comparison between test-time adaptation baselines and AdapT-able. We report the average adaptation time calculated over all test instances in CMC dataset corrupted by Gaussian noise.

**Table 5:** Post-training time analysis of uncertainty calibrator in AdapTable under different scales, encompassing small-scale (MLP + CMC), medium-scale (FT-Transformer + HELOC), and large-scale (TabNet + DIABETES READMISSION).

| Setting | Uncertainty Calibrator Training Time (s) |
|---|---|
| CMC, MLP | 4.46 |
| HELOC, FT-Transformer | 9.27 |
| DIABETES READMISSION, TabNet | 281.16 |

With previously calculated online target label estimator, we predict the debiased target label estimator $\boldsymbol{p}_t^{\mathrm{de}}(y|\boldsymbol{x}_i^t)$ and estimate the current target label distribution $\boldsymbol{p}_t(y)$ with

$$\boldsymbol{p}_t^{\mathrm{de}}(y|\boldsymbol{x}_i^t) = \sigma_{L_1}\Big(\boldsymbol{p}_t(y|\boldsymbol{x}_i^t)/\boldsymbol{p}_s(y)\Big)$$

$$\boldsymbol{p}_t(y) = (1-\alpha)\cdot\sum_{i=1}^{N}\boldsymbol{p}_t^{\mathrm{de}}(y|\boldsymbol{x}_i^t)/N + \alpha\cdot\boldsymbol{p}_t^{\mathrm{oe}}(y).$$

After that, we quantile relative uncertainty $\epsilon_i$ among $\{\epsilon_{i'}\}_{i'=1}^{N}$ within current batch, and perform temperature sharpening for certain samples, whereas we perform temperature smoothing for uncertain samples with $\boldsymbol{p}_t(y|\boldsymbol{x}_i^t)' = \sigma_{\mathrm{Softmax}}\big(c_i\cdot\mathcal{F}(\boldsymbol{x}_i^t|\theta)\big)$, where $c_i$ is defined as in Equation 1. Using Bayes' theorem, we adjust the predicted probability of each instance $\boldsymbol{x}_i$ as $\sigma_{L_1}\big(\boldsymbol{p}_t(y|\boldsymbol{x}_i^t)'\boldsymbol{p}_t(y)/\boldsymbol{p}_s(y)\big)$, and using self-ensembling (Gao et al., 2023), we get the final prediction of $\hat{\boldsymbol{p}}_i(y) = \boldsymbol{p}_t(y|\boldsymbol{x}_i^t)'/2 + \sigma_{L_1}\big(\boldsymbol{p}_t(y|\boldsymbol{x}_i^t)'\boldsymbol{p}_t(y)/\boldsymbol{p}_s(y)\big)/2$, and update online target label estimator as follows:

$$\hat{\boldsymbol{p}}_i(y) = \boldsymbol{p}_t(y|\boldsymbol{x}_i^t)'/2 + \sigma_{L_1}\big(\boldsymbol{p}_t(y|\boldsymbol{x}_i^t)'\boldsymbol{p}_t(y)/\boldsymbol{p}_s(y)\big)/2$$

$$\boldsymbol{p}_t^{oe} \leftarrow (1-\alpha)\cdot\sum_{i=1}^{N}\hat{\boldsymbol{p}}_i(y)/N + \alpha\cdot\boldsymbol{p}_t^{oe}.$$

# D FURTHER EXPERIMENTS

## D.1 COMPUTATIONAL EFFICIENCY

In order to show the efficiency of the proposed AdapTable, we perform a computational efficiency comparison between test-time adaptation baselines and AdapTable in Figure 6. The averaged adaptation time is calculated by averaging adaptation time over all test instances in CMC dataset corrupted by Gaussian noise. We find that the adaptation time of AdapTable ranks third among eight TTA methods, by showcasing its computational tractability. Furthermore, we observe that our approach requires significantly less adaptation time compared to TTA baselines such as TTT++ (Liu et al.,

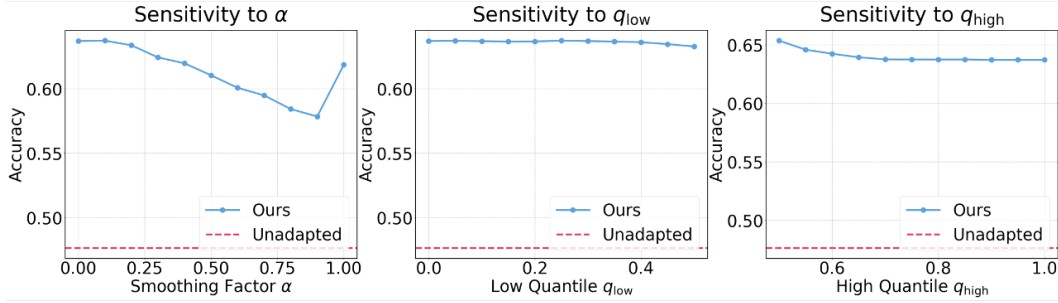

**Figure 7:** Hyperparameter sensitivity analysis of the proposed AdapTable using MLP under HELOC dataset with respect to smoothing factor $\alpha$, low uncertainty quantile $q_{\text{low}}$, and high uncertainty quantile $q_{\text{high}}$.

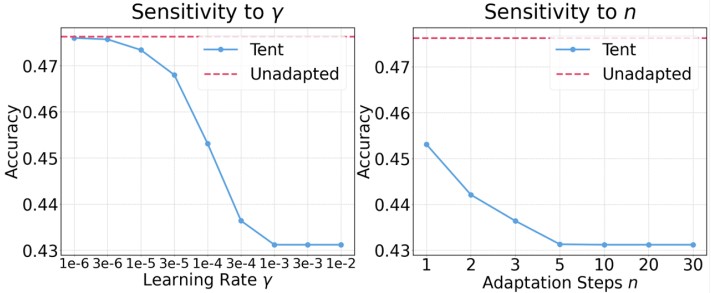

**Figure 8:** Hyperparameter sensitivity analysis of TENT (Wang et al., 2021a) using MLP under HELOC dataset with respect to learning rate $\gamma$, number of adaptation steps $n$.

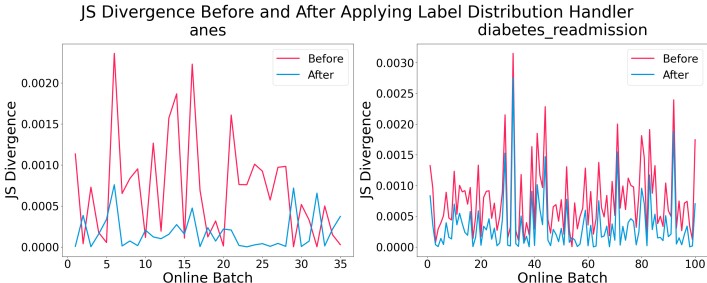

**Figure 9:** Jensen-Shannon Divergence between estimated target label distribution before and after applying label distribution handler upon ANES and DIABETES READMISSION dataset.

2021), SAM (Foret et al., 2021), EATA (Niu et al., 2022), and SAR (Niu et al., 2023), despite constructing shift-aware graph and incorporating a single forward pass for GNN with negligible extra cost of adjusting output label estimation are required. This can be attributed to the fact that the graph we generate places each column as a node, resulting in a graph of a very small scale – typically ranging from tens to hundreds of nodes. This minimizes the cost of message passing in GNN forward process, while other baselines iterate through multiple adaptation steps with forward and backward processes, leading to increased computational expenses. Furthermore, we also provide GNN post-training time of AdapTable under different scales, encompassing small-scale (CMC, MLP), medium-scale (HELOC, FT-Transformer), and large-scale (DIABETES READMISSION, TabNet). GNN post-training requires only a few seconds for small- and medium-scale settings, and notably, it remains negligible, even in our largest experimental setting.

## D.2    HYPERPARAMETER SENSITIVITY

To assess the robustness of AdapTable against hyperparameter configurations, we conduct an exhaustive hyperparameter sensitivity analysis covering all test-time parameters, including the smooth-

**Table 6:** Number of total columns, number of numerical and categorical columns along with their classes per datasets used.

| Property | HELOC | ANES | DIABETES READMISSION | CMC | MFEAT-PIXEL | DNA |
|---|---|---|---|---|---|---|
| # Columns | 22 | 54 | 46 | 9 | 240 | 180 |
| # Numerical | 20 | 8 | 12 | 2 | 240 | 0 |
| # Categorical | 2 | 46 | 34 | 7 | 0 | 180 |
| # Classes | 2 | 2 | 2 | 3 | 10 | 3 |

ing factor $\alpha$, low uncertainty quantile $q_{\text{low}}$, and high uncertainty quantile $q_{\text{high}}$. Specifically, in our experiments utilizing MLP on HELOC dataset, we perform hyperparameter optimization by varying one parameter while keeping the others fixed at the identified optimal setting, *i.e.*, $(\alpha, q_{\text{low}}, q_{\text{high}}) = (0.1, 0.25, 0.75)$. Notably, as Figure 7 exhibits, our findings reveal that the adaptation performance remains insensitive to alterations in all three types of hyperparameters, particularly when varying $q_{\text{low}}$, demonstrating minimal performance fluctuations. Furthermore, for the smoothing factor $\alpha$ and high quantile $q_{\text{high}}$, we pinpoint sweet spots at $[0, 0.2]$ and $[0.5, 0.6]$, respectively. This observation underscores the adaptability of our approach, allowing flexible hyperparameter selection and demonstrating generalizability across diverse test conditions. This stands in stark contrast to the hyperparameter sensitivity exhibited by the tent, as depicted in Figure 8. Notably, regardless of an extensive hyperparameter search in the tabular domain for the tent, the performance post-adaptation fails to surpass the unadapted performance, as evidenced by our main table experiment results in Table 2 and Table 3.

### D.3    EFFICACY OF LABEL DISTRIBUTION HANDLER

Figure 9 reports the Jensen-Shannon Divergence (JS Divergence) of each test batch between ground truth label distribution and prediction, comparing before and after adaptation using label distribution handler for ANES (Studies, 2022) and DIABETES READMISSION (Clore et al., 2014) datasets, further from Section 4.3. The figure consistently exhibits a decrease in divergence after adaptation, which solidifies the efficacy of the label distribution handler 3.3.

## E    DATASET DETAILS

### E.1    DATASETS

In our experiment, we verify our method across six different datasets. Among them, three datasets (HELOC, ANES, and DIABETES READMISSION) include natural distribution shifts between training and test data, while the other ones (CMC, MFEAT-PIXEL, and DNA) does not have such shifts, and thus we synthetically inject noises (Section E.2) on them to mimic plausible distribution shift scenarios. In our experiments, each dataset is partitioned as follows: 60% for training, 20% for validation, and 20% for testing. For all datasets, the numerical features are normalized – subtraction of mean and division by standard deviation, while categorical features are one-hot encoded. We find that different encoding types do not play a significant role in terms of accuracy, as noted in (Grinsztajn et al., 2022). Detailed specifications of each dataset are listed in Table 6

- HELOC: Home Equity Line of Credit (HELOC) (Brown et al., 2018) dataset is the dataset to predict whether the applicant will repay their HELOC account within two years; which is a line of credit typically offered by a bank as a percentage of home equity. Data is split with respect to external risk estimation value; lower ones are used for test data.

- ANES: American National Election Studies (ANES) (Studies, 2022) provide classification task of U.S. presidential election participation. Domain shift is given by the geographic region of surveyees.

- DIABETES READMISSION: Diabetes Readmission (Clore et al., 2014) represents ten years (1999-2008) of clinical care at 130 US hospitals and integrated delivery networks. Each row concerns hospital records of patients diagnosed with diabetes, who underwent laboratory, medications, and stayed up to 14 days. The goal is to determine the early readmission of the

patient within 30 days of discharge. Admission sources are different between train and test data.

- CMC: Contraceptive Method Choice (CMC) is a subset of Contraceptive Prevalence Survey conducted in Indonesia. The goal is to predict the current contraceptive method choice – between no-use, long-term methods, or short-term methods, with respect to the woman's demographic and socio-economic characteristics. The train and test data were split with respect to the most important column's values. Since it contains both numerical and categorical features, we obtained two different splits by selecting one most important column from the numerical columns, and one from the categorical columns.

- MFEAT-PIXEL: Multiple Features Dataset – Pixel (MFEAT-PIXEL) is a handwritten digit recognition dataset. Its goal is to classify handwritten numerals extracted from Dutch utility maps. The input is digitized, and all the pixel values are in binary form, 0 corresponding to black and 1 corresponding to white. The train and test data were split with respect to the most important column's values.

- DNA: Primate Splice-Junction Gene Sequences consist of splice junction of DNA, described by 180 indicator variables. The goal is to recognize the 3 classes – 1. boundaries between exons(retained after splicing), 2. introns(removed after splicing), or 3. none of the above. The dataset stems from Irvine database, but with major differences including the processing of symbolic variables representing the nucleotides, and the names of each example. The train and test data were split with respect to the most important column's values.

### E.2 Synthetic Corruptions

Let $\boldsymbol{x} = [x_1, \cdots, x_D]$ be a single table row with $D$ columns, where $\mu_i$ and $\sigma_i$ denote the mean and the standard deviation of the empirical marginal distribution of the $i$-th column calculated by the training set. We inject four synthetic corruptions to mimic aleatoric uncertainty, and two natural-shift oriented synthetic shifts to mimic natural distribution shifts.

- Gaussian Noise: For each column $x_i$, we add a Gaussian noise $\epsilon$ with $x_i \leftarrow x_i + \epsilon \cdot \sigma_i$ independently, where $\epsilon \sim \mathcal{N}(0, 0.1^2)$.

- Uniform Noise: For each column $x_i$, we add a uniform noise $\epsilon$ with $x_i \leftarrow x_i + \epsilon \cdot \sigma_i$ independently, where $\epsilon \sim \mathcal{U}(-0.1, 0.1)$.

- Random Missing: For each column $x_i$, we mask and replace it by using a random mask $m_i$ and a random sample $\bar{x}_i$ with $x_i \leftarrow (1 - m_i) \cdot x_i + m_i \cdot \bar{x}_i$, where $m_i \sim \text{Ber}(0.2)$, $P(\bar{x}_i = k) = \sum_{i=j}^{n_s} \mathbb{1}_{[\boldsymbol{x}_{j,i}^s = k]}/n_s$ for $k \in \mathbb{R}$. $n_s$ is the number of train instances, and $\boldsymbol{x}_{j,i}^s$ denotes the $i$-th column of the $j$-th train sample. We assume that we have knowledge of which columns are missing.

- Random Column Missing: This is similar to the random missing corruption, except for the fact that all test instances across multiple batches have the same common columns missing. For each column $x_i$, we mask and replace it with a random sample using a random mask $m_i$ and a random sample $\bar{x}_i$ with $x_i \leftarrow (1 - m_i) \cdot x_i + m_i \cdot \bar{x}_i$, where $m_i \sim \text{Ber}(0.2)$, $P(\bar{x}_i = k) = \sum_{i=j}^{n_s} \mathbb{1}_{[\boldsymbol{x}_{j,i}^s = k]}/n_s$ for $k \in \mathbb{R}$. $n_s$ is the number of train instances, and $\boldsymbol{x}_{j,i}^s$ denotes the $i$-th column of the $j$-th train sample. We also assume that we have knowledge of which columns are missing.

- Numerical Column Shift: This shift mimics natural domain shifts, we extract the most important numerical column based on pre-trained XGBoost (Chen & Guestrin, 2016) and sort all instances in the dataset according to the most important numerical column, and the top 80% of the data is predominantly allocated to the training and validation sets, while the lower 20% is primarily assigned to the test set.

- Categorical Column Shift: This shift also mimics natural domain shifts, we extract the most important categorical column based on pre-trained XGBoost (Chen & Guestrin, 2016) and split the train test dataset accordingly. Instances belonging to the category that is most frequently represented within the top 80% are predominantly assigned to the training and validation sets. Conversely, instances associated with the category that has the least frequent occurrences within the lower 20% are mainly allocated to the test set.

# F    BASELINE DETAILS

## F.1    DEEP TABULAR LEARNING ARCHITECTURES

- MLP: Multi-layer perceptron (MLP) (Murtagh, 1991): is a foundational deep learning architecture characterized by multiple layers of interconnected nodes, where each node applies a non-linear activation function to a weighted sum of its inputs. In the tabular domain, MLP is often employed as a default deep learning model, with each input feature corresponding to a node in the input layer.

- TabNet: TabNet (Arik & Pfister, 2021) introduces a unique blend of decision trees and neural networks. It utilizes an attention mechanism to selectively focus on informative features at each decision step, making it particularly well-suited for handling tabular data with a mix of categorical and continuous features.

- FT-Transformer: FT-Transformer (Gorishniy et al., 2021), short for feature tokenizer along with Transformer (Vaswani et al., 2017), represents a straightforward modification of the Transformer architecture tailored for tabular data. In this model, the feature tokenizer component plays a crucial role by converting all features, whether categorical or numerical, into tokens. Subsequently, a series of Transformer layers are applied to these tokens within the Transformer component, along with the added [CLS] token. The ultimate representation of the [CLS] token in the final Transformer layer is then utilized for the prediction.

## F.2    SUPERVISED BASELINES

- K-NN: k-Nearest Neighbors (k-NN) is a widely used model in tabular learning, that measures distance between data points using a chosen metric to identify its k-nearest neighbors, and makes predictions through majority voting for classification, or weighted averaging for regression. k is a user-defined hyperparameter, influencing the sensitivity of the model.

- LR: Logistic Regression (LR) is a linear classification algorithm for tabular data that models the probability of an instance belonging to a particular class. Using a logistic function to squash the linear combination of input features into a range of $[0, 1]$. With the appropriate regularization techniques, it has shown its capability to be comparable with SOTA architectures in the tabular domain.

- RF: Random Forest (RF) is an ensemble learning (bagging) algorithm that constructs multiple decision trees to enhance accuracy and mitigate overfitting. It excels in handling non-linear patterns, providing high accuracy and robustness against outliers.

- XGBOOST: Extreme Gradient Boosting (XGBoost) (Chen & Guestrin, 2016) is an ensemble learning (boosting) algorithm building a sequence of weak learners, usually decision trees, to correct errors of the previous model. It stands out for its high predictive performance, ability to handle complex relationships and regularization features.

- CATBOOST: CatBoost (Dorogush et al., 2017), similar to XGBoost, is a boosting ensemble algorithm. It efficiently handles categorical features without extensive pre-processing, making it advantageous for real-world datasets. Its benefits include high performance, but it comes at a computational cost. Additionally, parameter tuning may be necessary for optimal results.

## F.3    TEST-TIME ADAPTATION BASELINES

- PL: Pseudo-labeling (PL) (Lee, 2013) uses a pseudo-labeling strategy to update the model weights during test-time.

- TTT++: Test-time training (TTT++) (Liu et al., 2021) tries to mitigate deterioration of test-time adaptation performance through feature alignment strategies, regularizing the adaptation, without the need to re-access source data.

- TENT: Test entropy minimization (Tent) (Wang et al., 2021a) updates the scale and bias parameters within the batch normalization layer with entropy minimization during test-time, with a given test batch.

**Table 7:** Hyperparameter search space of supervised baselines. # Neighbors denotes the number of neighbors, # Estim denotes the number of estimators, Dep th denotes the maximum depth, and LR denotes the learning rate, respectively.

| Baseline | Search Space |
|---|---|
| K-NN | # Neighbors: {2 - 12} |
| RF | # Estim: {50 - 200}, Depth: {2 - 12} |
| XGBOOST | # Estim: {50 - 200}, Depth: {2 - 12}, LR: {0.01 - 1}, Gamma: {0 - 0.5} |
| CATBOOST | # Estim: {50 - 200}, Depth: {5 - 40} |

**Table 8:** Hyperparameter search space of test-time adaptation baselines. Here, we only denote the common hyperparameters, where method specific hyperparameters are specified in Section G.2.

| Hyperparameter | Search Space |
|---|---|
| Learning Rate | {1e-3, 1e-4, 1e-5, 1e-6} |
| Adaptation Steps | {1, 5, 10, 15, 20} |
| Episodic | {True, False} |

- SAM: Sharpness-aware minimization (SAM) (Foret et al., 2021) although not a method devised for test-time adaptation, has shown its effectiveness combined with TENT through updating parameters that lie in neighborhoods having uniformly low loss.

- EATA: Efficient Anti-forgetting Test-time Adaptation (EATA) (Niu et al., 2022) points out that samples with high entropy may lead to unreliable gradients that disrupt the model. EATA filters these high-entropy samples along with utilizing a fisher regularizer to constrain important model parameters during adaptation.

- SAR: Sharpness-aware and reliable optimization (SAR) (Niu et al., 2023) improves upon SAM, armed with the observation – samples with large entropy leads to model collapse during test-time, and filters the samples for adaptation with a pre-defined threshold.

- LAME: Laplacian adjusted maximum-likelihood estimation (LAME) (Boudiaf et al., 2022) is a new approach towards test-time adaptation, adapting without parameter optimization, but only corrects the output probabilities of a classifier rather than tweaking the model's inner parameters.

# G HYPERPARAMETER DETAILS

## G.1 SUPERVISED BASELINES

For k-nearest neighbors (K-NN), logistic regression (LR), random forest (RF), XGBOOST (Chen & Guestrin, 2016), and CATBOOST (Dorogush et al., 2017), optimal parameters are searched for each datasets using random search of 100 iterations, for each dataset. The search space for each method is specified in Table 7.

## G.2 TEST-TIME ADAPTATION BASELINES

Entropy minimization-based methods, namely TENT Wang et al. (2021a), SAM Foret et al. (2021), and SAR Niu et al. (2023), require 2 main hyperparameters – learning rate, number of adaptation steps per batch, and whether to reset the model after batch (*i.e.*, episodic adaptation). Additionally, SAR Niu et al. (2023) requires a threshold hyperparameter to filter samples with high entropy. For TENT, we set the learning rate as 0.0001 with 1 adaptation step and episodic update. For SAM Foret et al. (2021) and SAR Niu et al. (2023), the learning rate is 0.001 with 1 adaptation step and episodic update. For PL Lee (2013), we set the learning rate as 0.0001 with 1 adaptation step and episodic updates. For TTT++ (Liu et al., 2021), EATA (Niu et al., 2022) and LAME (Boudiaf et al., 2022), we find that the author's hyperparameter choices are optimal, as specified in their paper and official code, except for their learning rate and adaptation steps. For TTT++ (Liu et al., 2021) and EATA (Niu et al., 2022), the

**Table 9:** Selected hyperparameters of test-time adaptation baselines. In this table we only denote the common hyperparameters, where method specific hyperparameters are specified in text.

| Baseline | Learning Rate | Adaptation Steps | Episodic |
|----------|---------------|------------------|----------|
| PL       | 1e-4          | 1                | True     |
| TTT++    | 1e-5          | 10               | True     |
| TENT     | 1e-4          | 1                | True     |
| SAM      | 1e-3          | 1                | True     |
| EATA     | 1e-5          | 10               | True     |
| SAR      | 1e-3          | 1                | True     |
| LAME     | N/A           | N/A              | N/A      |

**Table 10:** Selected hyperparameters of AdapTable. Three major hyperparameters – smoothing factor $\alpha$, low quantile $q_{\text{low}}$, high quantile $q_{\text{high}}$ are specified below per architecture. The hyperparameters were fixed throughout datasets.

| Architecture   | $\alpha$ | $q_{\text{low}}$ | $q_{\text{high}}$ |
|----------------|----------|------------------|-------------------|
| MLP            | 0.1      | 0.25             | 0.75              |
| TabNet         | 0.0      | 0.25             | 0.9               |
| FT-Transformer | 0.0      | 0.25             | 0.9               |

learning rate is set to 0.00001 with 10 adaptation steps per batch and episodic updates. LAME Boudiaf et al. (2022) only corrects the output logits, thus not requiring hyperparameters related to gradient updates. For all previous baselines, we find that their hyperparameter choice did not vary across different architectures, namely MLP, TabNet (Arik & Pfister, 2021) and FT-Transformer (Gorishniy et al., 2021). As noted in the main paper, all hyperparameters for the corresponding method and backbone architecture pair are tuned with respect to numerical shift on CMC dataset from OpenML-CC18 (Bischl et al., 2021). An overview of the hyperparameter search space, along with selected hyperparameters of each method is provided in Table 8 and Table 9, respectively.

### G.3 ADAPTABLE

AdapTable requires three key test-time hyperparameters: smoothing factor $\alpha$, and low/high uncertainty quantiles $q_{\text{low}}/q_{\text{high}}$. The parameters for each backbone architecture are described in Table 10.

## H LIMITATIONS AND BROADER IMPACTS

### H.1 LIMITATIONS

Similar to other test-time training (TTT) methods, AdapTable incorporates an additional training procedure during the source model's training phase. This contrasts with fully test-time adaptation methods, which refrain from making assumptions during test-time execution. Specifically, AdapTable necessitates an extra post-training step for shift-aware uncertainty calibrator to adjust the model's predictions. While fully test-time adaptation methods, such as SAR (Niu et al., 2023), are applicable, their performance improvements in the tabular domain are limited, often failing to address certain covariate/label shifts, as evidenced in our evaluations. AdapTable demonstrates substantial performance gains in the majority of evaluation scenarios, although occasional shortcomings persist in specific datasets and model specifications.

### H.2 BROADER IMPACTS

Through comprehensive examinations, we have identified that the straightforward application of TTA methodologies from other domains, particularly those relying on entropy minimization, which currently constitutes the most prevalent form of TTA, encounters substantial challenges in the context of tabular data. Notably, these challenges arise from the high uncertainty of prediction entropy for

tabular data, potentially leading to model collapse, as exemplified in SAR (Niu et al., 2023), and the inadequacy of the cluster assumption within the latent space of models trained on tabular data. Moreover, most TTA methodologies are tailored exclusively to deep learning models, an assumption often overlooked in domains where deep learning models have surpassed traditional machine learning approaches. However, this assumption cannot be dismissed in the tabular domain, where classical machine learning methods, such as decision trees, form a competitive baseline.

In contrast, AdapTable, by refining only the output probabilities and circumventing noisy backpropagation from high-entropy-prone data, avoids failure. Additionally, our novel shift-aware uncertainty calibrator leverages the heterogeneous characteristics of columns, enabling our method to effectively address domain shift. We posit that our work empowers future researchers to adeptly confront the crucial challenge of mitigating domain shift in tabular data – an arena where the application of prior methods from other domains is not straightforward due to the aforementioned issues.

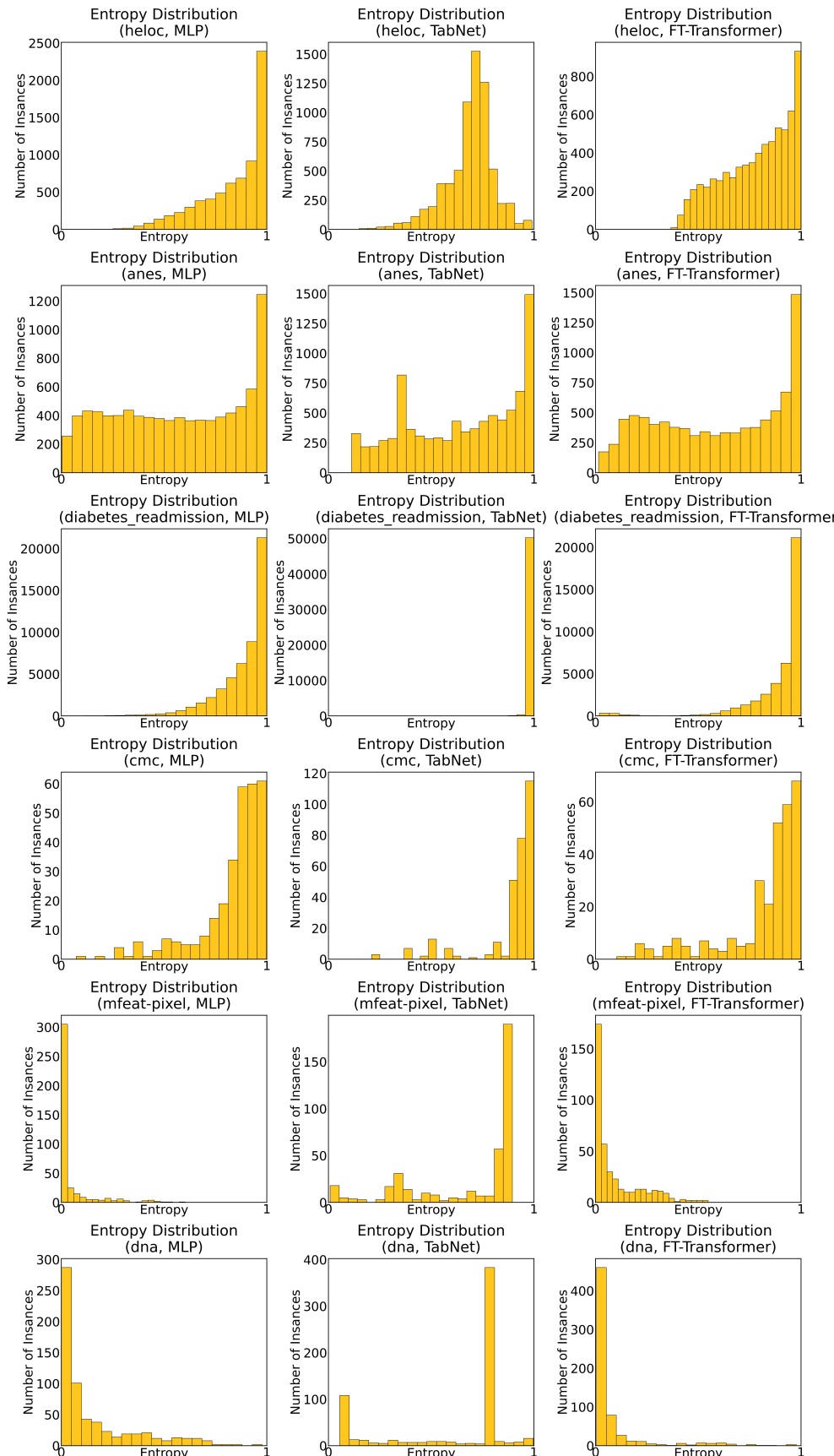

**Figure 10:** Entropy distribution histograms for test instances across six different datasets and three representative deep tabular learning architectures.

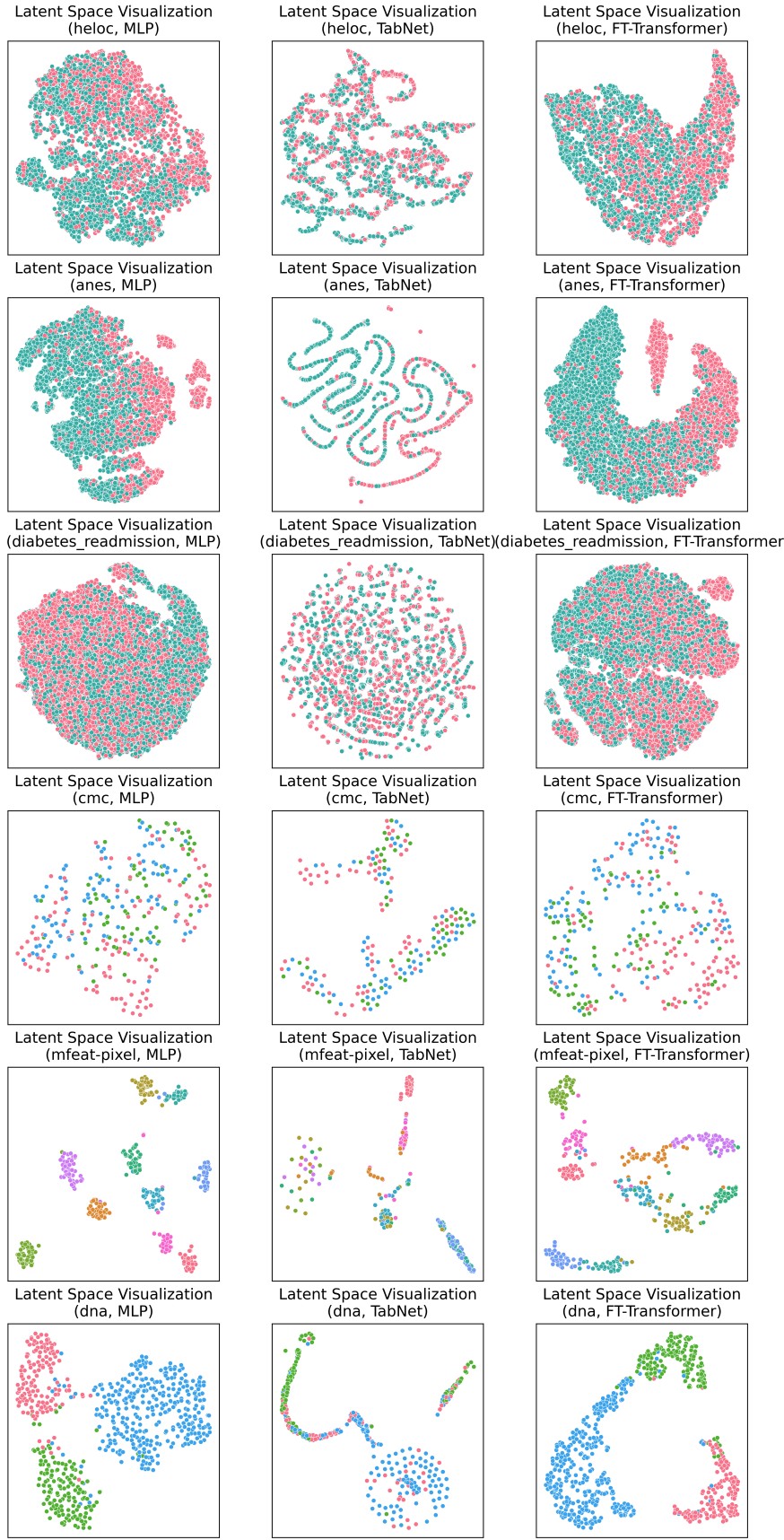

**Figure 11:** Latent space visualizations for test instances using t-SNE across six different datasets and three representative deep tabular learning architectures.

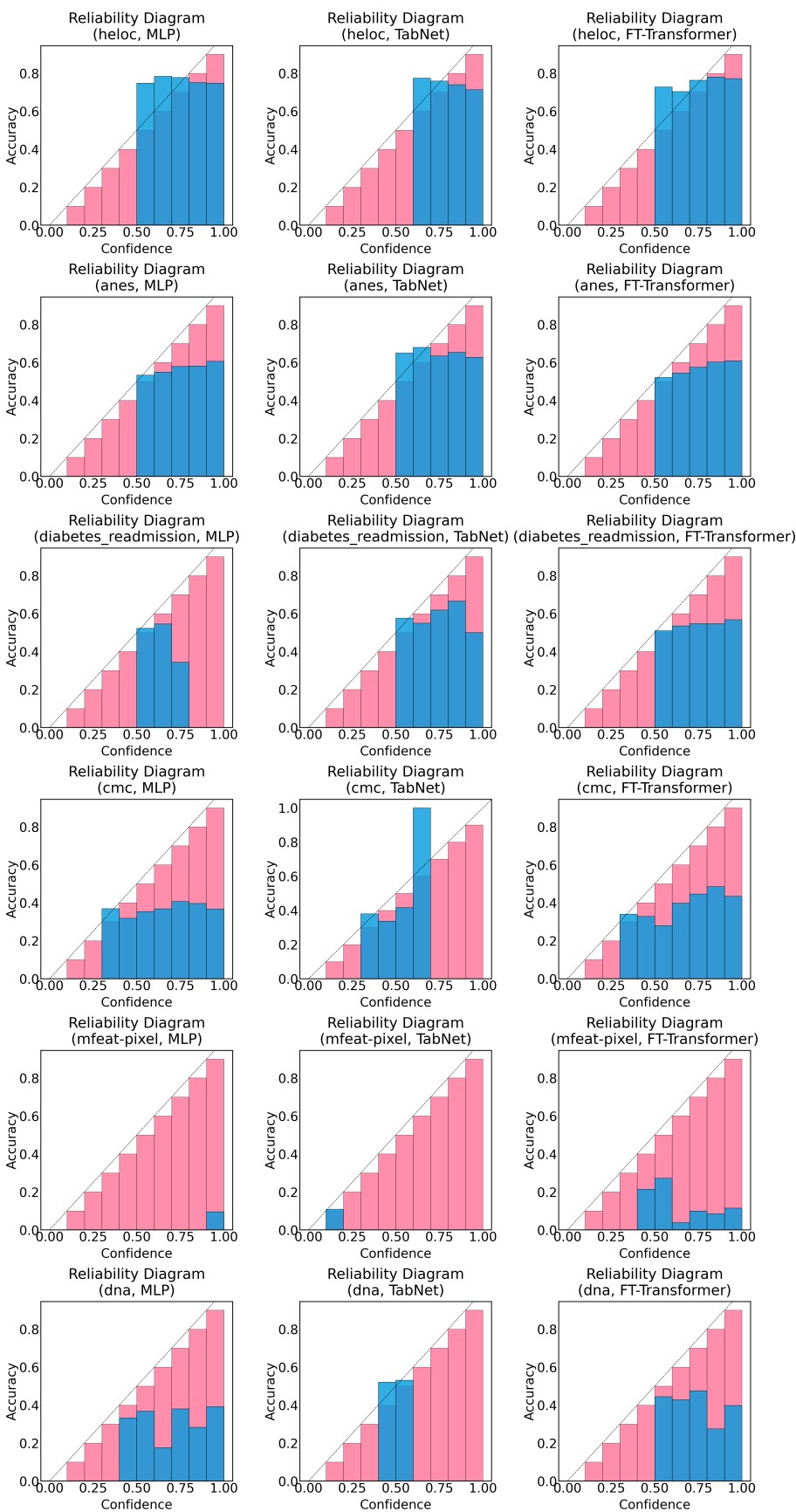

**Figure 12:** Reliability diagrams for test instances across six different datasets and three representative deep tabular learning architectures.

