# OpenReview forum: "AdapTable: Test-Time Adaptation for Tabular Data via Shift-Aware Uncertainty Calibrator and Label Distribution Handler"
_ICLR.cc/2024/Conference — Submitted to ICLR 2024_

### Official Review · Reviewer_DhMn · 2023-10-23

**Soundness:** 2 fair
**Presentation:** 3 good
**Contribution:** 2 fair
**Rating:** 5
**Confidence:** 4

**Summary:**

This paper proposes AdapTable to address problems about tabular-specific test-time adaption. AdapTable uses a shift-aware uncertainty calibration module to correct the poor confidence calibration, and uses a label distribution handler to adjust the output distribution. Experimental results show the effectiveness of AdapTable.

**Strengths:**

This paper comprehensively investigate the problems about tabular-specific test-time adaption, and propose a new method called AdapTable to address these problems.

**Weaknesses:**

This paper does not investigate and justify (semantically and experimentally) the advantages of the proposed post-hoc output calibration method compared with traditional model calibration methods such as isotonic regression calibration and Platt calibration. Besides, it is not clear how the target label distribution output by the calibration model is guaranteed to be correct.

**Questions:**

1. What are the advantages of the calibration method proposed in this paper compared to the traditional calibration method such as isotonic regression calibration and Platt calibration?
2. How is the target label distribution ensured (or ensured with a certain probability) to be correct?

---

> ### Author Response · Authors · 2023-11-17
> **Response to Reviewer DhMn [1]**
>
> We sincerely thank the reviewer for the helpful and insightful comments. We address the reviewer's concerns below.
>
> ### [W1, Q1. This paper does not investigate and justify (semantically and experimentally) the advantages of the proposed post-hoc output calibration method compared with traditional model calibration methods such as isotonic regression calibration and Platt calibration. Besides, it is not clear how the target label distribution output by the calibration model is guaranteed to be correct. / What are the advantages of the calibration method proposed in this paper compared to the traditional calibration method such as isotonic regression calibration and Platt calibration?]
>
> The calibration method proposed in the 'AdapTable' paper, referred to as the shift-aware post-hoc uncertainty calibrator, offers several advantages over traditional calibration methods like isotonic regression calibration and Platt calibration, especially in the context of handling distribution shifts in tabular data. Here are the key advantages:
>
> **Shift-Awareness:** One of the main innovations of the calibration method in 'AdapTable' is its awareness of distribution shifts. Traditional methods like isotonic regression and Platt calibration are typically applied in a setting where the training and test data are assumed to be drawn from the same distribution. In contrast, the shift-aware calibrator in **'AdapTable' is specifically designed to handle distribution shifts,** making it more suitable for real-world scenarios where the test data might differ significantly from the training data.
>
> **Graph Neural Network Integration:** The calibrator in 'AdapTable' utilizes a Graph Neural Network (GNN) to **understand the relationships between different features (columns) of the data.** This approach is particularly beneficial for tabular data, which often comprises a mix of numerical and categorical features, and each feature encompasses a distinctive meaning, unlike pixels within vision domains. Traditional methods do not typically leverage such relational information between features, which can be crucial for accurate calibration in complex datasets.
>
> **Handling Underconfidence and Overconfidence:** The 'AdapTable' method **addresses both underconfidence and overconfidence in predictions,** a significant improvement over traditional methods. Platt calibration, for instance, is often more focused on correcting overconfident predictions and may not be as effective in dealing with underconfidence.
>
> To address the concerns of the reviewer, we’ve additionally conducted an ablative study by exchanging the calibrator to a. Platt calibrator *(PC)* and b. Isotonic regression *(IR)* in the table below. It is empirically shown that such calibration methods such as Platt Calibration, and Isotonic Regression, denoted as their abbreviations, show suboptimal performance when jointly used with our label shift handler. (DIABETES indicates DIABETES READMISSION dataset.)
>
> | (Acc. %)  | HELOC      | ANES       | DIABETES   | CMC        | MFEAT-PIXEL | DNA        |
> | ---- | ---------- | ---------- | ---------- | ---------- | ----------- | ---------- |
> | Unadapted | 47.0 ±1.6 | 79.3 ±0.2 | 61.3 ±0.1| 53.5 ±1.4 | 96.4 ±0.2 | 91.4 ±1.2 |
> | PC   | 61.6 ± 1.3 | 79.2 ± 0.2 | 61.4 ± 0.3 | 52.1 ± 1.5 | 96.4 ± 0.3  | 91.4 ± 0.8 |
> | IR   | 61.3 ± 1.7 | 79.2 ± 0.2 | 61.0 ± 0.4 | 51.9 ± 1.6 | 96.0 ± 0.3  | 91.3 ± 0.8 |
> | **Ours** | **64.5 ± 0.3** | **79.6 ± 0.1** | **61.7 ± 0.0** | **55.7 ± 2.0** | **97.8 ± 0.2**  | **95.0 ± 0.5** |

---

> ### Author Response · Authors · 2023-11-17
> **Response to Reviewer DhMn [2]**
>
> ### [Q2. How is the target label distribution ensured (or ensured with a certain probability) to be correct?]
>
> Ensuring the accuracy of the target label distribution in our 'AdapTable' framework involves two key strategies:
>
> **De-biasing Source Label Distribution:** Grounded in Bayes' theorem, we address the inherent bias in the prediction logits of the source model towards the source label distribution. This is achieved by dividing the prediction logits by the source label distribution, which is obtained prior to the model's deployment. This step effectively debiases the predictions, making them more representative of the actual target distribution rather than being overly influenced by the source data characteristics.
>
> **Iterative Refinement Strategy:** The estimation and adjustment of the target label distribution in 'AdapTable' is an iterative process. As the model encounters more test data, it continually refines its understanding and estimation of the target label distribution. This iterative refinement is key to gradually improving the accuracy of the label distribution estimation, ensuring that the model adapts more effectively to the target domain over time.
>
> With these strategies, we are able to estimate the target label distribution with a considerable degree of accuracy. The empirical performance of this approach is demonstrated in our paper, particularly in **Figure 5** and **Figure 9** in our revised paper, where we plot the Jensen-Shannon divergence of distributions before and after applying our label distribution handler. This plot illustrates the effectiveness of our method in aligning the model’s predictions with the actual label distribution in the target domain, showcasing the practical utility of our approach in handling label distribution shifts in real-world scenarios.
>
> We sincerely appreciate the reviewer again for the thoughtful comments.

---

> ### Author Response · Authors · 2023-11-21
> **Thanks for the feedback! Have we adequately resolved the issues raised?**
>
> We express our gratitude for dedicating time to review our paper. In our rebuttal, we have expounded on the contribution of our work, offering a more comprehensive explanation and additional experiments for the proposed framework.
>
> Considering the limited duration of the author-reviewer discussion phase, we seek your input on whether our primary concerns have been sufficiently tackled. We are prepared to furnish further explanations and clarifications if needed. Thank you sincerely!

---

### Official Review · Reviewer_P5dW · 2023-10-29

**Soundness:** 3 good
**Presentation:** 4 excellent
**Contribution:** 2 fair
**Rating:** 5
**Confidence:** 4

**Summary:**

Tabular data in real-world applications often face distribution shifts, impacting machine learning model performance during testing. Addressing these shifts in tabular data is challenging due to varying attributes and dataset sizes, and limitations of deep learning models for tabular data. To tackle these challenges, the AdapTable method is introduced, which estimates target label distributions and adjusts initial probabilities based on calibrated uncertainty, demonstrating its effectiveness in experiments with real-world and synthetic data shifts using unlabeled test data alone.

**Strengths:**

1. TTT on tabular data has its unique challenge, and authors well clarify this point in Sec.2. I really appreciate such clarification.
2. Experimental results are extensive and convincing.

**Weaknesses:**

1. Why SHIFT-AWARE UNCERTAINTY CALIBRATOR and  LABEL DISTRIBUTION HANDLER are combined into each other? I mainly  concern on whether your solution looks like a A+B combination. If you can clarify this point, I will be pleased to raise my score.

**Questions:**

See Weaknesses.

---

> ### Author Response · Authors · 2023-11-17
> **Response to Reviewer P5dW**
>
> We sincerely thank the reviewer for the helpful and insightful comments. We address the reviewer's concerns below.
>
> ### [W1. Why SHIFT-AWARE UNCERTAINTY CALIBRATOR and LABEL DISTRIBUTION HANDLER are combined into each other? I mainly concern on whether your solution looks like a A+B combination. If you can clarify this point, I will be pleased to raise my score.]
>
> Thank you for your constructive feedback regarding the integration of the Shift-Aware Uncertainty Calibrator and the Label Distribution Handler in our **'AdapTable'** framework. We realize we may not have clearly articulated the rationale behind this integration, and we apologize for any confusion caused. To address this issue more comprehensively, we have added additional insights in **section 2.2** of our revised paper.
>
> The key points summarized in this section are as follows:
>
> **Pivotal Observation:** A critical observation we made is that when the source model is appropriately calibrated – increasing confidence for correct samples while decreasing it for incorrect ones – our Label Distribution Handler significantly enhances performance. This improvement is especially notable because it directly tackles the inherent challenge in tabular data, where label distribution shifts are common. This content can be found in Table 1 of our revised paper.
>
> **Challenges in Obtaining Good Calibration:** Achieving effective calibration in scenarios with distribution shifts is a complex task. To address this, we utilize Graph Neural Networks (GNNs) in our Shift-Aware Uncertainty Calibrator. The GNNs are adept at learning the impact of shifts in each column on the label, thus ensuring robust calibration even when distribution changes occur during test-time. This is crucial for our label-shift handler to show effectiveness even under distributional shifts such as: (1) distribution change of certain columns, (2) observation noises on certain features, etc.
>
> In summary, the combination of the Shift-Aware Uncertainty Calibrator and the Label Distribution Handler in 'AdapTable' is a synergistic approach where **each component complements and enhances the effectiveness of the other.** The calibration provided by the Shift-Aware Uncertainty Calibrator lays the groundwork for the Label Distribution Handler to make more precise adjustments, reflecting the actual label distribution shifts in the target domain. This integrated approach is designed to address the unique challenges posed by both covariate and label shifts in tabular data, providing a comprehensive solution for test-time adaptation.
>
> We sincerely appreciate the reviewer again for the thoughtful comments.

---

> ### Author Response · Authors · 2023-11-21
> **Thanks for the feedback! Have we adequately resolved the issues raised?**
>
> We express our gratitude for dedicating time to review our paper. In our rebuttal, we have expounded on the contribution of our work, offering a more comprehensive explanation and additional experiments for the proposed framework.
>
> Considering the limited duration of the author-reviewer discussion phase, we seek your input on whether our primary concerns have been sufficiently tackled. We are prepared to furnish further explanations and clarifications if needed. Thank you sincerely!

---

### Official Review · Reviewer_xrMn · 2023-10-31

**Soundness:** 2 fair
**Presentation:** 3 good
**Contribution:** 2 fair
**Rating:** 3
**Confidence:** 5

**Summary:**

This paper studies an interesting problem of test-time adaptation for tabular data. This problem is meaningful since tabular data suffers from distribution shift problems while lacking effectiveness in solving them. The authors propose a model-independent test-time adaptation method, which estimates the temperature for each sample during testing and modifies the label distribution of outputs. The experiments show a significant performance improvement on three datasets in the TableShift benchmarks.

**Strengths:**

1. This paper studies an interesting problem, namely test-time adaptation for tabular data.
2. The proposed method is suitable for addressing label distribution shifts in tabular datasets.

**Weaknesses:**

1. The experiments in this paper are insufficient. There are 15 datasets in the TableShift benchmark; however, only three of them are considered in the experiments. This makes the results of this paper unconvincing.
2. The performance improvement on ANES and Diabetes ReadMission is relatively weak in the Supervised setting (which is the setting with the overall best performance). This makes the proposed method weak.
3. The hyper-parameter of the proposed method lacks thoughtful discussion. For example, the alpha in the label distribution handler determines how quickly the model can adapt to the latest label distribution, which may seriously affect the performance.
4. This method calibrates the logits with sample-wise temperature and estimated label distribution, which can only handle the label distribution shift problems rather than covariate shift problems.
5. Minor issue: The left sub-figures in Figures 2 and 5 contain black borders.

**Questions:**

1. The authors should explain why only three datasets are adopted in the experiments and why these three datasets are selected.
2. The running time of the proposed method should be reported for both training and evaluation.
3. Please refer to the questions in the weakness.

---

> ### Author Response · Authors · 2023-11-17
> **Response to Reviewer xrMn [1]**
>
> We sincerely thank the reviewer for the helpful and insightful comments. We address the reviewer's concerns below.
>
> ### [W1, Q1. The experiments in this paper are insufficient. There are 15 datasets in the TableShift benchmark; however, only three of them are considered in the experiments. This makes the results of this paper unconvincing. / The authors should explain why only three datasets are adopted in the experiments and why these three datasets are selected.]
>
> Your concern regarding the selection of only three datasets from the TableShift benchmark for our experiments is indeed valid and warrants a detailed explanation. In 'AdapTable', our focus was on demonstrating the method's effectiveness in real-world scenarios, and the choice of datasets was guided by these objectives.
>
> **Representativeness:**
>
> The datasets HELOC, Diabetes Readmission, and ANES are foundational and widely recognized in their respective data science domains. Their significant citation counts – over 2000 for HELOC and DIABETES, and more than 8000 for ANES – are indicative of their broad acceptance and extensive utilization in research.
>
> **Realistic Scenario and Practical Relevance:**
>
> Test-time adaptation (TTA) is particularly pertinent when domain shifts are prevalent during the deployment stage of models, thus we selected datasets that target a realistic scenario in which (1) deployment of pretrained models, and (2) domain shift during test-time, is likely. Each of the selected datasets – HELOC, DIABETES, and ANES – represents a distinct and realistic application of machine learning in the fields of healthcare, finance, and political science, where domain shifts such as regions may directly affect the underlying distribution of data.
>
> **Additional Experiments for Broader Validation:**
>
> Acknowledging the potential limitation of using a relatively small number of datasets for validation, we have conducted additional experiments within the TableShift benchmark. These experiments include datasets varying in size, such as ASSISTments with 2,667,776 observations and Childhood Lead with 27,499 observations. For these experiments, we used the same backbone model (MLP) as in our initial studies. This extension of our experimental evaluation further demonstrates the adaptability and effectiveness of 'AdapTable' across a broader range of scenarios and dataset sizes.
>
> To verify the validity of ours in various datasets in TableShift, we conduct extra experiments in TableShift datasets (BRFSS BLOOD PRESSURE, NHANES LEAD, and ASSISMENTS) with MLP backbone, and the results are in the table in the following comment. We confirm that our method is not tailored for some specific TableShift datasets.

---

> ### Author Response · Authors · 2023-11-17
> **Response to Reviewer xrMn [2] (Table)**
>
> (1), (2), and (3) indicate dataset BRFSS BLOOD PRESSURE, NHANES LEAD, and ASSISMENTS each.
>
> |  | (1) |  |  | (2) |  |  | (3)|  |  |
> | --- | --- | --- | --- | --- | --- | --- | --- | --- | --- |
> |  | acc | bacc | f1 | acc | bacc | f1 | acc | bacc | f1 |
> | baseline | 53.9  | 49.5  | 47.2  | 92.2  | 50.0  | 48.0  | 58.0  | 62.8  | 54.1  |
> | PL | 54.0  | 49.4  | 46.8  | 92.2  | 50.0  | 48.0  | 58.0  | 62.8  | 54.1  |
> | EM | 49.0  | 47.8  | 47.8  | 92.2  | 50.0  | 48.0  | 58.3  | 63.0  | 54.6  |
> | SAM | 57.2  | 49.6  | 38.5  | 92.2  | 50.0  | 48.0  | 55.9 | 60.7 | 51.4 |
> | SAR | 45.7 | 43.2 | 41.1 | 92.2  | 50.0  | 48.0  | 58.3  | 63.0  | 54.5  |
> | TTT++ | 51.1 | 50.5 | 50.4 | 92.2  | 50.0  | 48.0  | 58.3  | 63.0  | 54.5  |
> | EATA | 49.4  | 51.1  | 49.4  | 92.2  | 50.0  | 48.0  | 58.4  | 63.1  | 54.7 |
> | LAME | 44.6 | 50.3 | 38.0 | 92.2  | 50.0  | 48.0  | 58.3  | 63.0  | 54.5  |
> | Ours | **59.8** | **62.3** | **59.6** | **93.3** | **58.6** | **56.4** | **58.6** | **63.8** | **54.9** |

---

> ### Author Response · Authors · 2023-11-17
> **Response to Reviewer xrMn [3]**
>
> ### [W2. The performance improvement on ANES and Diabetes ReadMission is relatively weak in the Supervised setting (which is the setting with the overall best performance). This makes the proposed method weak.]
>
> As discernible from the main paper text or algorithm names, architectures labeled 'supervised' in the Table 1 and Table 2 are currently not adopted in other modalities but are still embraced in the tabular domain due to their performance, representing relatively classical algorithms that diverge from modern deep learning practices. Furthermore, a comparison focusing solely on the highest performances across the entire table indicates that these methods, as commonly known, continue to be adopted due to their ability to surpass deep learning performance (regardless of the architecture of deep learning models). When dealing with tabular data in a data-wise manner, many issues arise from this, the difficulty in utilizing recent deep learning-based research analyses and their derivatives, including test time adaptation. However, our approach differs from other TTA methods by not requiring updates or gradients from the main model. Our method is capable of orthogonal utilization even with algorithms that are SoTA but do not seamlessly align with deep learning methodologies. Despite its amount, this has resulted in performance improvements in numerous cases, demonstrating that using our method alongside classical algorithms is not only feasible but also a rational choice.
>
> ### [W3. The hyper-parameter of the proposed method lacks thoughtful discussion. For example, the alpha in the label distribution handler determines how quickly the model can adapt to the latest label distribution, which may seriously affect the performance.]
>
> To assess the robustness of AdapTable against hyperparameter configurations, we conduct an exhaustive hyperparameter sensitivity analysis covering all test-time parameters, including the smoothing factor $\alpha$, low uncertainty quantile $q_{\text{low}}$, and high uncertainty quantile $q_{\text{high}}$. Specifically, in our experiments utilizing MLP on HELOC dataset, we perform hyperparameter optimization by varying one parameter while keeping the others fixed at the identified optimal setting, *i.e.*, $(\alpha, q_{\text{low}}, q_{\text{high}}) = (0.1, 0.25, 0.75)$. Notably, as Figure 7 exhibits, our findings reveal that the adaptation performance remains insensitive to alterations in all three types of hyperparameters, particularly when varying $q_{\text{low}}$, demonstrating minimal performance fluctuations. Furthermore, for the smoothing factor $\alpha$ and high quantile $q_{\text{high}}$, we pinpoint sweet spots at $[0, 0.2]$ and $[0.5, 0.6]$, respectively. This observation underscores the adaptability of our approach, allowing flexible hyperparameter selection and demonstrating generalizability across diverse test conditions. This stands in stark contrast to the hyperparameter sensitivity exhibited by the tent, as depicted in Figure 8 in our revised appendix of paper. Notably, regardless of an extensive hyperparameter search in the tabular domain for the tent, the performance post-adaptation fails to surpass the unadapted performance, as evidenced by our main table experiment results in Table 2 and Table 3.
>
> ### [W4. This method calibrates the logits with sample-wise temperature and estimated label distribution, which can only handle the label distribution shift problems rather than covariate shift problems.]
>
> While our approach does not explicitly target covariate shifts, **the nature of the shift in tabular data differs from that in visual data, where a specific corruption can easily be applied to samples while preserving class information.**
>
> In most cases in tabular data, a shift in input space is highly correlated with the shift in the target label  - as shown in the table in the following comment. For instance, in a tabular dataset concerning patient and health information, a shift in the age range of patients directly affects other features and output labels. Our method also deals with covariate shifts by modeling unobserved shifts through the interdependence of shift values during shift-aware calibration. Additionally, by leveraging GNNs to learn the dependence of each column’s shifts, our approach demonstrates superior performance even in synthetic shift scenarios such as random noise or column missing, which are label-shift-free, highlighting its effectiveness in handling covariate shifts.
>
> ### [W5. Minor issue: The left sub-figures in Figures 2 and 5 contain black borders.]
>
> We sincerely thank you for letting us resolve this issue. We update Figure 2 and Figure 5 without black borders in our new draft.

---

> ### Author Response · Authors · 2023-11-17
> **Response to Reviewer xrMn [4] (Table)**
>
> | (%)                  |       | Class 0 | Class 1 |
> | -------------------- | ----- | ------- | ------- |
> | **HELOC**                | Train | 75.9    | 24.1    |
> || Valid                | 80.2  | 19.8    |
> || Test                 | 43.1  | 56.9    |
> | **ANES**                 | Train | 31.8    | 68.2    |
> || Valid                | 32.3  | 67.7    |
> || Test                 | 40.1  | 59.9    |
> | **DIABETES READMISSION** | Train | 57.6    | 42.4    |
> || Valid                | 57.8  | 42.2    |
> || Test                 | 50.6  | 49.4    |
> | **BRFSS BLOOD PRESSURE** | Train | 59.7    | 40.3    |
> || Valid                | 60.1  | 39.9    |
> || Test                 | 41.9  | 58.1    |
> | **NHANES LEAD**          | Train | 97.3    | 2.7     |
> || Valid                | 96.6  | 3.4     |
> || Test                 | 92.2  | 7.8     |
> | **ASSISMENTS**           | Train | 30.6    | 69.4    |
> || Valid                | 30.7  | 69.3    |
> || Test                 | 56.3  | 43.7    |

---

> ### Author Response · Authors · 2023-11-17
> **Response to Reviewer xrMn [5]**
>
> ### [Q2. The running time of the proposed method should be reported for both training and evaluation.]
>
> In order to show the efficiency of the proposed AdapTable, we perform a computational efficiency comparison between test-time adaptation baselines and AdapTable in **Figure 6**. The averaged adaptation time is calculated by averaging adaptation time over all test instances in CMC dataset corrupted by Gaussian noise. We find that the adaptation time of AdapTable ranks third among eight TTA methods, by showcasing its computational tractability. Furthermore, we observe that our approach requires significantly less adaptation time compared to TTA baselines such as TTT++[1], SAM[2], EATA[3], and SAR[4], despite constructing shift-aware graph and incorporating a single forward pass for GNN with negligible extra cost of adjusting output label estimation are required. This can be attributed to the fact that the graph we generate places each column as a node, resulting in a graph of a very small scale -- typically ranging from tens to hundreds of nodes. This minimizes the cost of message passing in GNN forward process, while other baselines iterate through multiple adaptation steps with forward and backward processes, leading to increased computational expenses.
>
> Furthermore, we also provide GNN post-training time of AdapTable under different scales in Table 5, encompassing small-scale (CMC, MLP), medium-scale (HELOC, FT-Transformer[5]), and large-scale (Diabetes Readmission, TabNet[6]). GNN post-training requires only a few seconds for small- and medium-scale settings, and notably, it remains negligible, even in our largest experimental setting.
>
> We sincerely appreciate the reviewer again for the thoughtful comments.
>
>
> [1] Yuejiang Liu, et al. "TTT++: When does self-supervised test-time training fail or thrive?" NeurIPS 2021.
>
> [2] Pierre Foret, et al. "Sharpness-aware minimization for efficiently improving generalization" ICLR 2021.
>
> [3] Niu, Wu, Zhang, et al. "Efficient test-time model adaptation without forgetting" ICML 2022.
>
> [4] Niu, Wu, Zhang, et al. "Towards stable test-time adaptation in dynamic wild world" ICLR 2023.
>
> [5] Yury Gorishniy, et al. "Revisiting deep learning models for tabular data" NeurIPS 2021.
>
> [6] Sercan Ö Arik, et al. "TabNet: Attentive interpretable tabular learning" AAAI 2021.

---

> ### Author Response · Authors · 2023-11-21
> **Thanks for the feedback! Have we adequately resolved the issues raised?**
>
> We express our gratitude for dedicating time to review our paper. In our rebuttal, we have expounded on the contribution of our work, offering a more comprehensive explanation and additional experiments for the proposed framework.
>
> Considering the limited duration of the author-reviewer discussion phase, we seek your input on whether our primary concerns have been sufficiently tackled. We are prepared to furnish further explanations and clarifications if needed. Thank you sincerely!

---

### Official Review · Reviewer_A2qG · 2023-11-01

**Soundness:** 3 good
**Presentation:** 3 good
**Contribution:** 2 fair
**Rating:** 5
**Confidence:** 4

**Summary:**

This paper focuses on the test-time adaptation problem for the tabular data. Specifically, the authors discuss the challenges related to test-time adaptation for tabular data and propose a new method including two modules: a shift-aware uncertainty calibration module and a label distribution handler. Experimental results show the proposal can improve learning performance on various datasets.

**Strengths:**

1) This paper studies the problem of adaptation for tabular data. This problem is important yet under-studied.

2) The authors conduct a large number of experiments and the results show that the proposal can improve performance.

**Weaknesses:**

1) Although the proposal can improve the performance, it is mainly a combination of some existing techniques. There is neither good theoretical analysis nor much inspiration, thus, the novelty and contribution are limited. From my personal point of view, I don’t really appreciate papers that improve performance by integrating multiple tricks and existing methods.
2) For tabular data, the shift may exist in various perspectives, such as the distribution shift, the feature dimension (new features occur or old features are lost), and class space varies. These problems should discussed separately.
3) Will the introduction of GCN in the method lead to a higher computational complexity of the algorithm? This should be discussed theoretically or empirically.

**Questions:**

As discussed above.

---

> ### Author Response · Authors · 2023-11-17
> **Response to Reviewer A2qG [1]**
>
> We sincerely thank the reviewer for the helpful and insightful comments. We address the reviewer's concerns below.
>
> ### [W1. Although the proposal can improve the performance, it is mainly a combination of some existing techniques. There is neither good theoretical analysis nor much inspiration, thus, the novelty and contribution are limited. From my personal point of view, I don’t really appreciate papers that improve performance by integrating multiple tricks and existing methods.]
>
> We conducted extensive experiments to understand the characteristics of deep tabular learned models and the features of domain shifts in the tabular data. Our findings can be summarized as follows:
> 1) Deep-learned models for tabular data are often observed to be uncalibrated in terms of uncertainty, exhibiting an overall underconfident nature, in contrast to the widely recognized overconfident behavior of vision domains. (This discovery, especially considering that prior techniques in TTA and domain adaptation predominantly stemmed from vision-based research with many early methods relying on smoothness assumptions like EM and pseudo-labeling, underscores the value of this work.)
> 2) Tabular domain shifts, unlike in the vision domain as well, affect the actual output label in most cases, resulting in discernible differences in label distributions between training and testing. While methods addressing label distribution shifts in vision domains have emerged recently, the task itself simulates situations where the output label distribution is adjusted without the cases that the shifts affect the output label, as seen in scenarios like Cifar-corrupted or ImageNet-corrupted datasets. It is crucial to emphasize the distinct nature of tabular data, where shifts in input (covariate) directly influence label and label distribution shifts, prompting the proposal of observation-based, tabular-specific modules capable of handling these shifts.
> 3) When utilizing label distribution shift handlers in deep learning models, the effectiveness of the label distribution handler amplifies when the output of the deep learning model is well-calibrated in terms of uncertainty, which is a relatively well-known fact than previous ones. This statement holds in tabular settings, where this effect appears to be substantial. In our revised paper, **Table 1** indicates that if the source model is perfectly calibrated by increasing the confidence for correct samples while decreasing the confidence for incorrect samples, our label distribution handler leads to a remarkable improvement in performance. These interconnected observations emphasize the need for a calibrator tailored for tabular data and a unique handler to address shifts specific to the tabular domain. This elucidates the essential interdependence between both in addressing the nuanced shifts observed in tabular datasets.
>
> ### [W2. For tabular data, the shift may exist in various perspectives, such as the distribution shift, the feature dimension (new features occur or old features are lost), and class space varies. These problems should be discussed separately.]
>
> Like many previous test-time adaptation (TTA) works, our primary focus in the 'AdapTable' paper is on handling distribution shifts. These shifts pertain to changes in the data distribution between the training (source) and testing (target) phases, without affecting the input dimension or the number of features. This is a critical point of distinction, as our method is designed under the assumption that the input features (feature space) and the output classes (class space) remain fixed throughout.
>
> Specifically, our method is designed to handle column-wise distributional shifts(corresponding to splitting training and test data with respect to the distribution of important columns), but has shown its effectiveness in handling other shifts as well such as noise(gaussian and random noise), and missing data (random drop, column drop), as demonstrated in our experimental section. In which all cases do not incur dimensional changes nor additional classes during testing. It should be noted that for missing features, imputation was done to match the feature dimensions of the input.
>
> In summary, **AdapTable** is designed to address distributional shifts within tabular data where the dimensionality of input features and number of output classes remain constant - aligning with previous works in TTA.

---

> ### Author Response · Authors · 2023-11-17
> **Response to Reviewer A2qG [2]**
>
> ### [W3. Will the introduction of GCN in the method lead to a higher computational complexity of the algorithm? This should be discussed theoretically or empirically.]
>
> In order to show the efficiency of the proposed AdapTable, we perform a computational efficiency comparison between test-time adaptation baselines and AdapTable in **Figure 6**. The averaged adaptation time is calculated by averaging adaptation time over all test instances in CMC dataset corrupted by Gaussian noise. We find that the adaptation time of AdapTable ranks third among eight TTA methods, by showcasing its computational tractability. Furthermore, we observe that our approach requires significantly less adaptation time compared to TTA baselines such as TTT++ [1], SAM [2], EATA [3], and SAR [4], despite constructing shift-aware graph and incorporating a single forward pass for GNN with negligible extra cost of adjusting output label estimation are required. This can be attributed to the fact that the graph we generate places each column as a node, resulting in a graph of a very small scale -- typically ranging from tens to hundreds of nodes. This minimizes the cost of message passing in GNN forward process, while other baselines iterate through multiple adaptation steps with forward and backward processes, leading to increased computational expenses.
>
> Furthermore, we also provide GNN post-training time of AdapTable under different scales in Table 5, encompassing small-scale (CMC, MLP), medium-scale (HELOC, FT-Transformer [5]), and large-scale (Diabetes Readmission, TabNet [6]). GNN post-training requires only a few seconds for small- and medium-scale settings, and notably, it remains negligible, even in our largest experimental setting.
>
> We sincerely appreciate the reviewer again for the thoughtful comments.
>
>
> [1] Yuejiang Liu, et al. "TTT++: When does self-supervised test-time training fail or thrive?" NeurIPS 2021.
>
> [2] Pierre Foret, et al. "Sharpness-aware minimization for efficiently improving generalization" ICLR 2021.
>
> [3] Niu, Wu, Zhang, et al. "Efficient test-time model adaptation without forgetting" ICML 2022.
>
> [4] Niu, Wu, Zhang, et al. "Towards stable test-time adaptation in dynamic wild world" ICLR 2023.
>
> [5] Yury Gorishniy, et al. "Revisiting deep learning models for tabular data" NeurIPS 2021.
>
> [6] Sercan Ö Arik, et al. "TabNet: Attentive interpretable tabular learning" AAAI 2021.

---

> ### Author Response · Authors · 2023-11-21
> **Thanks for the feedback! Have we adequately resolved the issues raised?**
>
> We express our gratitude for dedicating time to review our paper. In our rebuttal, we have expounded on the contribution of our work, offering a more comprehensive explanation and additional experiments for the proposed framework.
>
> Considering the limited duration of the author-reviewer discussion phase, we seek your input on whether our primary concerns have been sufficiently tackled. We are prepared to furnish further explanations and clarifications if needed. Thank you sincerely!

---

### Author Response · Authors · 2023-11-17
**General Response**

Dear reviewers and AC,

We sincerely appreciate the dedication of time and effort in reviewing our manuscript.

As highlighted by the reviewers, we believe that our paper studies an interesting(xrMn) yet understudied(A2qG) problem of test-time adaptation on tabular data, which in itself proposes a unique challenge(P5dW), and through comprehensive analysis(DhMn) and extensive experiments(A2qG, P5dW) show the efficacy of our proposed method.

We appreciate all your constructive feedback,
We appreciate your helpful suggestions on our manuscript. In accordance with your comments, we have revised our manuscript with the following additional discussions and experiments:
* **Enhanced Clarity on Methodology**: We have provided a more detailed explanation in Section 2.2 on how the SHIFT-AWARE UNCERTAINTY CALIBRATOR and LABEL DISTRIBUTION HANDLER are jointly utilized in our method, offering a clearer understanding of their synergistic function.
* **Computational Efficiency Analysis**: Figures 6 in Appendix D.1 now present a comparative analysis of the computational efficiency between various test-time adaptation baselines and our **AdapTable** approach.
* **Hyperparameter Sensitivity Analysis**: Figure 7, 8 in Appendix D.2 offers an in-depth comparison of hyperparameter sensitivity between the test-time adaptation baselines and **AdapTable**, highlighting the robustness of our method.
* **Extended Evaluation of Label Distribution Handler**: Figure 9 in Appendix D.3 showcases the effectiveness of our Label Distribution Handler across additional datasets, empirically evidencing it accurate estimation of target label distribution.
* **Additional Observations and Analysis**: We have included more observations related to the latent space and entropy distributions in Appendix B - across different , providing deeper insights into the underlying mechanisms of our approach.

We thank all the reviewers once more for their valuable, constructive feedback.

Sincerely,

Authors.

---

### Author Response · Authors · 2023-11-21
**A Gentle Reminder**

Dear Reviewers and AC,

Thank you for your time and dedication again in reviewing our paper.

We kindly remind that the discussion period will end soon (in a few days).

We are confident that we have thoroughly and effectively addressed your comments, supported by the results of additional experiments.

If you have any remaining concerns or questions, please feel free to let us know.

Best regards,

Authors

---

### Author Response · Authors · 2023-11-23
**A Gentle Reminder for Final Hours of Discussion**

Dear Reviewers and AC,

We express our sincere gratitude once again, for your valuable time and dedicated effort in reviewing our paper.

As the discussion period draws to a close, we want to bring to your attention that there are now only few hours remaining before the deadline.

We want to assure you that we have diligently addressed all your comments and supplemented our responses with the outcomes of additional experiments, reinforcing the robustness of our work.

If you have any lingering concerns or queries, please do not hesitate to reach out to us within the next few hours. Your insights are pivotal to the refinement of our paper, and we highly appreciate your thorough review.

Thank you once again for your commitment to this process.

Best wishes,

Authors

---

### Author Response · Authors · 2023-11-23
**Final Reminder**

Dear Reviewers and AC,

Thank you for your time and effort in reviewing our paper.

As the discussion period nears its end, we'd like to highlight that only a few hours remain before the deadline.

We assure you that we've addressed your comments and included additional experiment results.

If you have any remaining concerns, please reach out in the next few hours. Your insights are crucial to us.

Thank you for your commitment.

Sincerely,

Authors

---

### Meta-Review · Area_Chair_DLoG · 2023-12-05

**Metareview:**

This work proposes a test-time adaptation strategy for tabular data.  Reviewers pointed out the Incremental conceptual contributions and limited empirical evaluation on insufficiently many datasets.  The number of datasets in this paper is far lower than other tabular papers, and this paper did not use an existing benchmark so that it is not clear whether or not the included datasets were cherry picked.  Given the strong performance shown on the three datasets in Table 2, I encourage the authors to run their experiments on a large suite.  This work has the potential to be impactful if the performance benefits do transfer broadly, so I encourage the authors to develop their work further.

**Justification For Why Not Higher Score:**

The evaluations are insufficient for the tabular domain, and the methodological contributions are not significant.

**Justification For Why Not Lower Score:**

N/A

---

### Decision · Program_Chairs · 2024-01-16

Reject